# PICProp: Physics-Informed Confidence Propagation for Uncertainty Quantification

**Qianli Shen**
National University of Singapore
Singapore
shenqianli@u.nus.edu

**Wai Hoh Tang**
National University of Singapore
Singapore
waihoh.tang@nus.edu.sg

**Zhun Deng**
Columbia University
USA
zhun.d@columbia.edu

**Apostolos Psaros**
Brown University
USA
a.psaros@brown.edu

**Kenji Kawaguchi**
National University of Singapore
Singapore
kenji@nus.edu.sg

## Abstract

Standard approaches for uncertainty quantification in deep learning and physics-informed learning have persistent limitations. Indicatively, strong assumptions regarding the data likelihood are required, the performance highly depends on the selection of priors, and the posterior can be sampled only approximately, which leads to poor approximations because of the associated computational cost. This paper introduces and studies confidence interval (CI) estimation for deterministic partial differential equations as a novel problem. That is, to propagate confidence, in the form of CIs, from data locations to the entire domain with probabilistic guarantees. We propose a method, termed Physics-Informed Confidence Propagation (PICProp), based on bi-level optimization to compute a valid CI without making heavy assumptions. We provide a theorem regarding the validity of our method, and computational experiments, where the focus is on physics-informed learning. Code is available at https://github.com/ShenQianli/PICProp.

## 1 Introduction

Combining data and physics rules using deep learning for solving differential equations [25] and learning operator mappings [20] has recently attracted increased research interest. In this context, physics-informed machine learning is transforming the computational science field in an unprecedented manner, by solving ill-posed problems, involving noisy and multi-fidelity data as well as missing functional terms, which could not be tackled before [16]. Nevertheless, uncertainty – mainly due to data noise and neural network over-parametrization – must be considered for using such models safely in critical applications [24]. Despite extensive work on uncertainty quantification (UQ) for deep learning [1], standard UQ has persistent limitations. Indicatively, although for most practical applications where only confidence intervals (CIs) are required, the harder problem of UQ is often solved first using simplifying assumptions, and the CIs are subsequently computed. Further, strong assumptions regarding the data likelihood are required, the performance highly depends on

37th Conference on Neural Information Processing Systems (NeurIPS 2023).

the selection of priors, and the posterior can be sampled only approximately, which leads to poor approximations because of the associated computational cost.

In this paper, we introduce and study CI estimation for addressing deterministic problems involving partial differential equations (PDEs) as a novel problem. That is, to propagate confidence, in the form of CIs, from data locations to the entire domain with probabilistic guarantees. We name our framework Physics-Informed Confidence Propagation (PICProp) and propose a practical method based on bi-level optimization to identify valid CIs in physics-informed deep learning. Our method can be applied in diverse practical scenarios, such as propagating data CIs, either estimated or given by experts, to the entire problem domain without distributional assumptions. Indicative applications are provided in Appendix A.

PICProp provides distinct advantages over traditional UQ approaches in the context of physics-informed deep learning. Specifically, PICProp does not rely on strong assumptions regarding the data likelihood and constructs theoretically valid CIs. Further, PICProp offers a unique capability to construct joint CIs for continuous spatio-temporal domains. In contrast, UQ methods typically construct marginal intervals for discrete locations. Furthermore, due to the conservative propagation of uncertainty, PICProp produces conservative CIs, which is preferred in critical and risk-sensitive applications [7, 13], considering the significant harm of non-valid confidence intervals. This sets it apart from commonly used UQ methods like deep ensembles [18] and Monte-Carlo dropout [28, 10], which are known to be overconfident in practice [9, 7].

The paper is organized as follows: In Section 2 (and its Appendices B & C), we introduce the PDE problem setup and discuss related work. In Section 3 (and its Appendices D-F), we introduce the problem of CI estimation for PDEs and extensively describe joint solution predictions. Next, we formulate this novel problem as a bi-level optimization problem in Section 4 and Appendices G & H, and adopt various algorithms for computing the required implicit gradients in Appendix I. Finally, we evaluate our method on various forward deterministic PDE problems in Section 5 and Appendices J-L, and discuss our findings in Section 6.

## 2 Background

### 2.1 Problem Setup

Consider the following general PDE describing a physical or biological system:

$$\mathcal{F}[u(x), \lambda(x)] = 0, x \in \Omega, \qquad \mathcal{B}[u(x), \lambda(x)] = b(x), x \in \Gamma, \tag{1}$$

where $x$ is the $D_x$-dimensional space-time coordinate, $\Omega$ is a bounded domain with boundary $\Gamma$, and $\mathcal{F}$ as well as $\mathcal{B}$ are general differential operators acting on $\Omega$ and $\Gamma$, respectively. The PDE coefficients are denoted by $\lambda(x)$, the boundary term by $b(x)$, and the solution by $u(x)$.

In this paper, we focus on the data-driven forward deterministic PDE problem, where (i) operators $\mathcal{F}, \mathcal{B}$ and coefficients $\lambda$ are deterministic and known; (ii) datasets $\mathcal{D}_f = \{x_f^i\}_{i=1}^{N_f}$ for the force condition and $\mathcal{D}_b = \{(x_b^j, b^j)\}_{j=1}^{N_b}$ for the boundary condition are given; (iii) uncertainty only occurs in the boundary term due to data noise, i.e., $b^j = b(x_b^j) + \epsilon^j$ for $j = 1, ..., N_b$, and the rest of the dataset is considered to be deterministic without loss of generality. We denote the function obtained by utilizing the dataset $z = \{\mathcal{D}_f, \mathcal{D}_b\}$ for approximating the solution $u(x)$ of Equation (1) with a data-driven solver $\mathcal{A}$ as $u_{\mathcal{A}}(x; z)$.

Clearly, different datasets $z$ contaminated by random noise lead to different solutions $u(x; z)$ and thus, to a distribution of solutions at each $x$. It follows that the solution $u(x; Z)$ can be construed as stochastic, where $Z$ is used to denote the dataset as a random variable and $z$ to denote an instance of $Z$. In this regard, this paper can be viewed as an effort towards a universal framework, with minimal assumptions, for constructing CIs for solutions of nonlinear PDEs.

### 2.2 Related Work

**Physics-Informed Neural Networks** The physics-informed neural network (PINN) method, developed by [25], addresses the forward deterministic PDE problem of Equation (1) by constructing a neural network (NN) approximator $u_\theta(x)$, parameterized by $\theta$, for modeling the approximate solution

$u(x; z)$. The approximator $u_\theta(x)$ is substituted into Equation (1) via automatic differentiation for producing $f_\theta = \mathcal{F}[u_\theta, \lambda]$ and $b_\theta = \mathcal{B}[u_\theta, \lambda]$. Finally, the optimization problem for obtaining $\theta$ in the forward problem scenario given a dataset $z$ is cast as

$$\hat{\theta} = \operatorname*{argmin}_\theta \mathcal{L}_{pinn}(\theta, z), \quad \text{where} \quad L_{pinn} := \frac{w_f}{N_f} \sum_{i=1}^{N_f} ||f_\theta(x_f^i)||_2^2 + \frac{w_b}{N_b} \sum_{i=1}^{N_b} ||b_\theta(x_b^j) - b^j||_2^2, \quad (2)$$

and $\{w_f, w_b\}$ are objective function weights for balancing the various terms in Equation (2); see [23]. Following the determination of $\hat{\theta}$, $u_{\hat{\theta}}$ can be evaluated at any $x$ in $\Omega$. PINN can be scalled up for high-dimensional PDEs [14].

**Uncertainty Quantification for Physics-Informed Learning**  Although UQ for physics-informed learning is not extensively studied, the field is gaining research attention in recent years. Indicatively, [37] employed dropout to quantify the uncertainty of NNs in approximating the modal functions of stochastic differential equations; [36] presented a model based on generative adversarial networks (GANs) for quantifying and propagating uncertainty in systems governed by PDEs; [35] developed a new class of physics-informed GANs to solve forward, inverse, and mixed stochastic problems; [11] further studied Wasserstein GANs for UQ in solutions of PDEs; [34] proposed Bayesian PINNs for solving forward and inverse problems described by PDEs with noisy data. Although this field is always evolving, we refer to it in this paper as "standard" UQ in order to distinguish it from our proposed method in Section 4. In Appendices B-C, we summarize the limitations of standard UQ, and we direct interested readers to the review paper [24] for more information.

**Interval Predictor Model (IPM)**  The Interval Predictor Model (IPM) in regression analysis has a similar motivation as our approach. A function of interest is approximated by an IPM and the bounds on the function are obtained. [26] proposed a technique using a double-loop Monte Carlo algorithm to propagate a CI using the IPM. The technique requires sampling epistemic and aleatory parameters from uniform distributions to estimate the upper and lower bounds of the IPM. Our approach is different for two reasons. First, we fit a PINN model directly and thus, sampling epistemic parameters may not be feasible. Second, we do not require a sampling procedure. Instead, we cast the CI estimation as a bi-level optimization problem and solve it using hyper-gradient methods.

## 3   Confidence Intervals for Solutions of Nonlinear Partial Differential Equations

In this section, we provide the definition of CI for the exact PDE solution and introduce our main assumption.

**Definition 3.1** (Joint CI for PDE Solution).  A pair of functions $(L_Z, U_Z) \in \mathbb{R}^\Omega \times \mathbb{R}^\Omega$ is a $p$-CI for the exact PDE solution $u(x)$ if the random event $\mathcal{E} := \{\{L_Z(x) \leq u(x) \leq U_Z(x), \forall x \in \Omega\}$ holds with probability at least $p$.

Note that this interval refers to the exact solution $u(x)$. Therefore, a novel problem is posed in Definition 3.1, pertaining to propagating randomness, in the form of CIs, from the data locations, typically on the boundary, to the rest of the domain. The subscript $Z$ in $(L_Z, U_Z)$ signifies that the CI is random as it depends on the random dataset $Z$. The available data can be an instance of $Z$, denoted as $z$, or a collection of instances; see Appendix D for the considered problem scenarios. Such instances can, for example, be drawn from repeating independent experiments. The subscript $Z$ in $(L_Z, U_Z)$ is dropped if there is no ambiguity.

We also note that the joint CI (also referred to as confidence sequence [17, 32]) in Definition 3.1 corresponds to joint predictions $u(\cdot)$ over the whole domain $\Omega$, rather than to marginal predictions $u(x)$ for a given $x$. The importance of joint predictions as an essential component in UQ has been emphasized in recent works for active learning [30], sequential decision making [33], and supervised learning [22]. Nevertheless, the present paper is the first work to consider joint predictions, and general CI estimation, in the context of physics-informed learning. In Appendix E, we provide several theoretical remarks delineating the differences between marginal and joint predictions, as well as unions of bounds, which are typically used in standard UQ. In Appendix F, we highlight these differences with a toy example.

Next, to relate the PINN solution with the exact PDE solution $u(x)$ of Definition 3.1, we make the following assumption regarding the noise at the boundary.

**Assumption 3.2** (Zero-mean Noise). The noise on the boundary is zero-mean, i.e., $\mathbb{E}[\epsilon^j] = 0, \forall j \in \{1, \ldots, N_b\}$.

Next, a clean dataset (without noise), denoted as $\bar{z}$, can be considered as the expectation of $Z$, i.e., $\bar{z} = \mathbb{E}[Z]$. In this regard, consider a data-driven PDE solver $\mathcal{A}$, which yields an approximate solution $u_{\mathcal{A}}(x; z)$ given a dataset $z$. We define the properness of $\mathcal{A}$ based on the concept of a clean dataset.

**Definition 3.3** (Proper PDE Solver). A PDE solver $\mathcal{A}$ is $\eta$-proper with respect to a clean dataset $\bar{z}$, if $\max_{x \in \Omega} |u_{\mathcal{A}}(x; \bar{z}) - u(x)| \leq \eta$.

Henceforth, we only consider proper PDE solvers. That is, a powerful solver is available such that the absolute error of the solution obtained by the solver with a clean dataset is bounded. In all cases, we assume that such an informative clean dataset exists, but is not available. Further, the constant $\eta$, which is only related to $\bar{z}$ and $\mathcal{A}$, is considered known and fixed in the computational experiments. Note that no assumptions are made about $u(x; z)$ for $z \neq \bar{z}$, i.e., there are no assumptions about the PINN solver robustness against noisy data. Finally, in all NN training sessions we consider no NN parameter uncertainty, although it is straightforward to account for it by combining our method with Deep Ensembles [18], for instance. The aim of this paper is to propose a principled and theoretically guaranteed solution approach and thus, we refrain from applying such ad hoc solutions in the computational experiments and plan to consider parameter uncertainty rigorously in a future work. Nevertheless, we provide in Appendix L a comparison with Bayesian PINNs for completeness.

# 4 Method

In this section, we introduce our PICProp method and provide the accompanying theorem, as well as practical implementation algorithms. Further, we discuss the computational cost of our method and propose a modification for increasing efficiency. Specifically, based on Definition 3.1, we address the problem of propagating the uncertainty from the boundary to arbitrary locations $x \in \Omega$ of the domain. Equivalently, this relates to obtaining the corresponding $p$-CIs, i.e., obtaining the functions $(L, U) \in \mathbb{R}^{\Omega} \times \mathbb{R}^{\Omega}$ in Definition 3.1.

## 4.1 Physics-Informed Confidence Propagation (PICProp)

Our method comprises two steps: (i) a $p$-CI $\tilde{\mathcal{Z}}_p$ for the clean data $\bar{z}$ is constructed. (ii) $\tilde{\mathcal{Z}}_p$ is propagated to any point $x \in \Omega$ of the domain by solving the following problem,

$$L(x) = \min_{z \in \tilde{\mathcal{Z}}_p} u_{\mathcal{A}}(x; z) - \eta, \quad U(x) = \max_{z \in \tilde{\mathcal{Z}}_p} u_{\mathcal{A}}(x; z) + \eta, \tag{3}$$

which searches for configurations of points $z$ within the interval $\tilde{\mathcal{Z}}_p$ such that the predicted values for $u(x)$ given by the $\eta$-proper solver $\mathcal{A}$ are minimized or maximized, respectively.

The CI in the first step can either be given or constructed from data. The former case relates to various practical scenarios (e.g., the error ranges of measurement sensors with a certain confidence degree are known or given by experts), whereas the latter serves as a data-driven scheme, where less prior knowledge is required. Techniques to construct the CI in the first step depend on the data (noise) distribution. For example, if the noise is assumed to be Gaussian, the corresponding CI can be identified by chi-squared or Hotelling's $T$-squared statistics. If the noise distribution is not known, a concentration inequality can be used to identify the CI. Relevant details are provided in Appendix D. See also Figure 1 for an illustration.

The interval constructed via Equation (3) is a joint $p$-CI for the whole domain $\Omega$ of $u$. This fact is formalized in the next theorem (see Appendix G for the corresponding proof).

**Theorem 4.1.** *Consider a $p$-CI for $\bar{z}$ denoted as $\tilde{\mathcal{Z}}_p$, constructed such that $Pr(\bar{z} \in \tilde{\mathcal{Z}}_p) \geq p$. Then the functions $(L, U)$ obtained by solving Equation (3) define a $p$-CI for $u$.*

Although the robustness of the PDE solver does not affect the validity of Theorem 4.1, it nevertheless affects the quality of intervals constructed via Equation (3). Specifically, although valid CIs can always be constructed with arbitrary proper PDE solvers by solving Equation (3), a robust PDE solver less sensitive to data noise is expected to yield tighter CIs compared to a less robust solver.

Note that the confidence guarantee of the constructed CIs corresponds to a probability that is greater than, rather than equal to, $p$, which means that the constructed CIs are conservative. From a design perspective, conservative uncertainty estimation is the preferred route for critical and risk-sensitive applications [7, 13], although it can be more costly.

Next, a brute force and computationally expensive approach to solve Equation (3) pertains to traversing $z \in \tilde{\mathcal{Z}}_p$ in an exhaustive search (ES) manner and invoking a PDE solver for each candidate $z$. Alternatively, we propose to first employ a NN for obtaining $u_{\hat{\theta}(z)}$ by minimizing the physics-informed loss of Equation (2) and approximating $u(x; z)$ for each $z \in \tilde{\mathcal{Z}}_p$. In this regard, if a PINN is assumed to be an $\eta$-proper PDE solver, Equation (3) can be further extended to a bi-level optimization problem cast as

$$
L(x) = \min_{z \in \tilde{\mathcal{Z}}_p} u_{\hat{\theta}(z)}(x) - \eta, \quad U(x) = \max_{z \in \tilde{\mathcal{Z}}_p} u_{\hat{\theta}(z)}(x) + \eta,
$$
$$
s.t. \quad \hat{\theta}(z) = \arg\min_{\theta} \mathcal{L}_{pinn}(\theta, z).
$$
(4)

Nevertheless, explicit gradient $\nabla_z u$ calculation is intractable, as multiple steps of gradient descent for the NN are required for a reliable approximation. Despite this fact, bi-level optimization involving large-scale parameterized models such as NNs, with main applications in meta-learning, has been extensively studied in recent years and has resulted to numerous implicit gradient approximation methods (see Appendix I). In this regard, Algorithm 1 in Appendix H summarizes our method for solving efficiently Equation (3) by using such approximations. In passing, note that $\eta = 0$ is considered for all implementations in this work.

## 4.2 Efficient Physics-Informed Confidence Propagation (EffiPICProp)

If joint CIs are required for multiple query points, denoted by the set $\mathcal{X}_q$, Algorithm 1 has to be used multiple times to solve Equation (4) for each and every point $x$ in the set. This clearly incurs a high computational cost. To address this issue, a straightforward approach pertains to training an auxiliary model parameterized by $\psi$ that models the CI as $(L_\psi(x_q), U_\psi(x_q))$ for all query points $x_q \in \mathcal{X}_q$, or more generally of the entire domain $\Omega$. However, this naive regression using solutions of the bi-level optimization at a set of locations, which we refer to as SimPICProp, does not provide accurate results when the size of the query set is small; see computational results in Section 5.1.

For improved generalization properties with limited query points, we propose to train a two-input parameterized meta-model $u_\psi(x_q, x)$ that models the resulting PDE solution $u(x)$, via Equation (4), for each different query point. For example, note that $L(x)$ in Equation (4) evaluates the NN $u_{\hat{\theta}(z)}$ at the location of the query point and $z$ minimizes the corresponding value. As a result, if a trained model $u_\psi(x_q, x)$ minimizes the value of $u$ at $x_q$, the sought $p$-CI can be simply obtained by evaluating $u_\psi$ at $(x_q, x_q)$. More generally, given a query set consisting of $K$ randomly selected query points $\mathcal{X}_q = \{x_q^k\}_{k=1}^K$, the model is trained such that it solves the problem

$$
\min_\psi \frac{1}{2K} \sum_{k=1}^K \mathcal{L}_{pinn}(u_\psi^L(x_q^k, \cdot), z^L(x_q^k)) + \mathcal{L}_{pinn}(u_\psi^U(x_q^k, \cdot), z^U(x_q^k)),
$$
$$
\text{where} \quad z^L(x_q^k) = \arg\min_{z \in \tilde{\mathcal{Z}}_p} u_{\hat{\theta}(z)}(x_q^k), \; z^U(x_q^k) = \arg\max_{z \in \tilde{\mathcal{Z}}_p} u_{\hat{\theta}(z)}(x_q^k),
$$
$$
s.t. \quad \hat{\theta}(z) = \arg\min_{\theta} \mathcal{L}_{pinn}(u_\theta(\cdot), z).
$$
(5)

The estimated $p$-CI is, subsequently, given by

$$
L(x) = u_\psi^L(x, x), \quad U(x) = u_\psi^U(x, x).
$$
(6)

In fact, we can train a single model for both upper- and lower-bound predictions because the parameters are shared as in Equation (5). This is achieved by encoding an indicator into the input vector, so that $u_\psi^L(x_q, \cdot) = u_\psi(x_q, \cdot, -1)$ and $u_\psi^U(x_q, \cdot) = u_\psi(x_q, \cdot, 1)$. Furthermore, to balance the weights of the terms in the objective of Equation (5) more effectively, PINNs are separately trained with boundary conditions $z^L(x_q^{1:K})$ and $z^U(x_q^{1:K})$. In this regard, $u_\psi$ is obtained by

$$\min_{\psi} \mathcal{L}_{\text{Effi}} \left( x_q^{1:K}; \psi \right), \quad \text{where} \quad \mathcal{L}_{\text{Effi}} := \frac{1}{2K} \sum_{k=1}^{K} (1 - \lambda) \mathcal{L}(x_q^k, x_q^k; \psi) + \lambda \underset{x \in \Omega}{\mathbb{E}} \mathcal{L}(x_q^k, x; \psi), \tag{7}$$

$$s.t. \quad \mathcal{L}(x_q^k, x; \psi) := (u_\psi(x_q^k, x, -1) - u_{\hat{\theta}(z^L(x_q^k))}(x))^2 + (u_\psi(x_q^k, x, 1) - u_{\hat{\theta}(z^U(x_q^k))}(x))^2,$$

and $\lambda \in [0, 1]$ is a weight to balance the two loss terms, of which the first aims to enforce close predictions of CIs at query points, and the second to utilize extra information in PDE solutions for better generalization. We refer to this model as EffiPICProp and provide the corresponding implementation in Algorithm 2 in Appendix H. Specifically, the EffiPICProp version with $\lambda = 0$ is called SimPICProp, as it simply focuses on close predictions of confidence intervals at query points, while disregarding additional information at other locations.

## 5    Computational experiments

We provide three computational experiments that are relevant to physics-informed machine learning. We will start with a 1D pedagogical example to illustrate our proposed approaches. Next, we will move to more complicated problems involving standard representative equations met in many real-world problems including heat transfer, fluid dynamics, flows through porous media, and even epidemiology [21, 38]. Details regarding implementation, query points, and computational times can be found in Appendix J.

### 5.1    A Pedagogical Example

We consider the following equation, which has been adopted from the pedagogical example of [11],

$$u_{xx} - u^2 u_x = f(x), \quad x \in [-1, 1], \quad \text{(PDE)}, \tag{8}$$

where $f(x) = -\pi^2 \sin(\pi x) - \pi \cos(\pi x) \sin^2(\pi x)$. For the standard BCs $u(-1) = u(1) = 0$, the exact solution is $u(x) = \sin(\pi x)$. The purpose of this one-dimensional PDE with only two boundary points, i.e., with a two-dimensional search space in Equation (4), is to demonstrate the applicability of our method in different data scenarios (see Appendix D). Further, the small size of the problem is ideal for comparing computationally expensive methods, such as exhaustive search (ES) and employing PICProp separately at all domain locations, with the more efficient implementation of Algorithm 2.

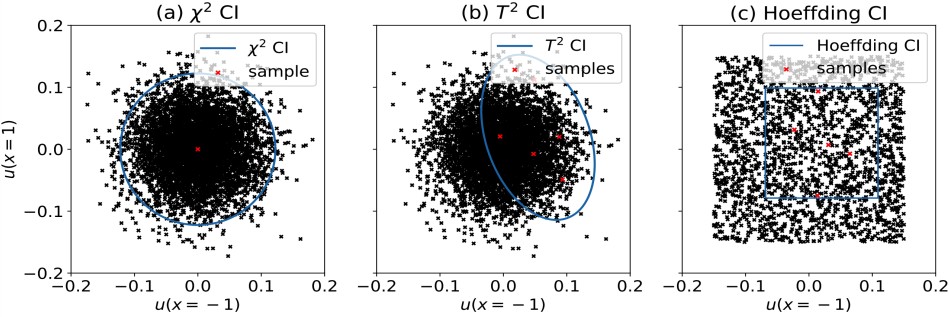

Figure 1: **Pedagogical example:** Boundary $95\%$ CIs to be propagated to the rest of the domain. (a-b) Random samples follow $\mathcal{N}(\mathbf{0}, \sigma^2 \mathbf{I})$, where $\sigma = 0.05$. The black crosses correspond to 5,000 random samples from the distribution. A $\chi^2$ CI is constructed using the only one available sample and a $T^2$ CI is constructed using the five available samples. (c) Random samples follow $\mathcal{U}([-1.5, 1.5] \times [-1.5, 1.5])$. The black crosses correspond to 2,000 random samples from the distribution. A Hoeffding CI is constructed using the five available samples. All available samples are shown with red crosses.

We consider two types of noisy BCs (Figure 1). First, we consider BCs with Gaussian noise given as

$$u(-1) \sim \mathcal{N}(0, \sigma_1^2), \quad u(1) \sim \mathcal{N}(0, \sigma_2^2), \tag{9}$$

where the standard deviations are $\sigma_1 = \sigma_2 = 0.05$, i.e., the data is assumed to be Gaussian and the standard deviations are known (or estimated if unknown). We sample one or five datapoint(s) for each side of the boundary, for each case, respectively. Given the dataset, a $\chi^2$ CI and a $T^2$ CI are constructed according to the corresponding problem scenarios discussed in Appendix D; see also Figure 1(a),(b).

Second, we consider BCs with uniform noise given as

$$u(-1) \sim \mathcal{U}(-D_1, D_1), \quad u(1) \sim \mathcal{U}(-D_2, D_2), \tag{10}$$

where $D_1 = D_2 = 1.5$ are assumed to be known. We sample five datapoints for each boundary side. Given the dataset, a Hoeffding CI is constructed; see also Figure 1(c).

For the query set, six points are used for implementing SimPICProp and EffiPICProp in Section 4.2, whereas 41 points are used for implementing the brute-force version of PICProp, which amounts to solving the bi-level optimization problem of Equation (4) for each point. The implementation details of the PICProp-based methods are summarized in Table J.1. Further, 5,000 BCs are sampled for solving Equation (4) with ES, which can be considered as a reference solution.

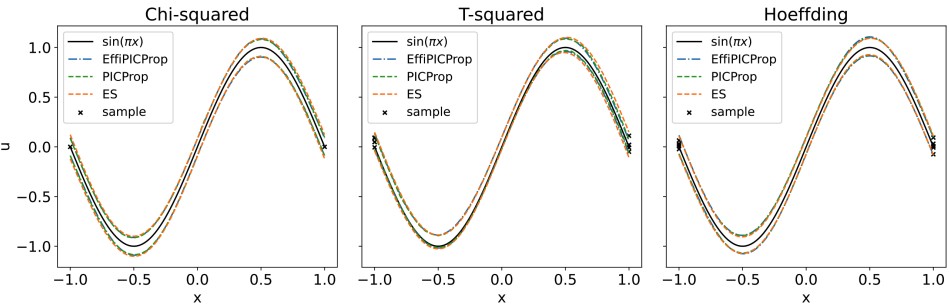

Figure 2: **Pedagogical example:** CI predictions for the entire domain $x \in [-1, 1]$. Left: Noise is Gaussian with known standard deviation, and a $\chi^2$ CI is constructed at the boundary. Middle: Noise is Gaussian with unknown standard deviation, and a $T^2$ CI is constructed at the boundary. Right: Uniform noise with known bounds, and a Hoeffding CI is constructed at the boundary.

The CI predictions for the three cases are visualized in Figure 2, whereas the meta-learning curves can be found in Appendix K.1. Note that the sizes of the CIs are approximately equal to $0.1 - 0.2$ near the boundaries, which are comparable with the propagated CI sizes of Figure 1, and become larger around $x$ of $\pm 0.5$. In all cases, the PICProp results match those of ES, despite the fact that PICProp requires solving a challenging bi-level optimization method with approximate implicit gradient methods. Further, the efficient method EffiPICProp performs accurately as compared to its reference method PICProp, while retaining a significantly lower computational cost (see Figure 3 for $\chi^2$ results and Figure K.4 K.5 for $T^2$ and Hoeffding results). On the other hand, the predictions of SimPICProp match the brute-force PICProp results only on the query points, but generalize poorly on other locations. In contrast, EffiPICProp, which is based on a meta-model that fits the PDE solutions for multiple query points, generalizes well on unseen locations.

Some additional experimental results are summarized in Appendix K.1 for the completeness of our empirical study: Firstly, a comparison among PICProp and EffiPICProp with various $\lambda$s is provided in Figure K.3 to better demonstrate how the balance between two EffiPICProp loss terms in Equation (7) affects the CI prediction. Next, we conduct an additional experiment to empirically verify the validity of EffiPICProp CIs in Appendix K.1.4. Finally, a brief comparison with Bayesian PINNs is presented in Appendix L.

## 5.2 Two-dimensional Poisson equation

Next, we consider a two-dimensional Poisson equation given as

$$
\begin{aligned}
u_{xx} + u_{yy} &= f(x, y), \quad (x, y) \in [-1, 1] \times [-1, 1], \quad \text{(PDE)}, \\
u(x, y) &= e^x + e^y, \quad x = \pm 1 \text{ or } y = \pm 1, \quad \text{(BCs)}.
\end{aligned}
\tag{11}
$$

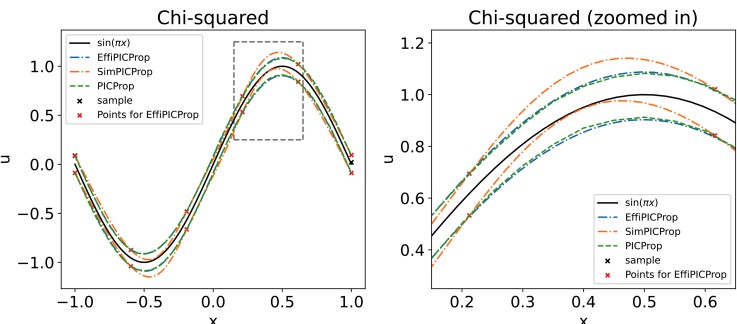

Figure 3: **Pedagogical example**, $\chi^2$: Although the results of both SimPICProp and EffiPICProp match PICProp results on the query points, EffiPICProp generalizes better on unseen locations.

The exact solution is $u(x, y) = e^x + e^y$, and the corresponding $f(x, y)$ is obtained by substituting $u(x, y)$ into Equation (11). This example serves to demonstrate the effectiveness of our method for treating cases with large numbers of noisy BC datapoints. In such cases, an exhaustive search approach for solving the bi-level optimization of Algorithm 1 generally fails to identify the global optimum, unless we increase the number of trials and the computational cost at prohibitive levels.

We propagate the following fixed CI from the boundary to the rest of the domain $(x, y) \in (-1, 1) \times (-1, 1)$:

$$u(x, y) \in [e^x + e^y - 0.05, e^x + e^y + 0.05], \quad x = \pm 1 \text{ or } y = \pm 1. \tag{12}$$

To solve the problem, 10 boundary points are sampled for each side of the domain, i.e., 40 boundary points in total are used for training and uncertainty is considered at all points. The search space is thus 40-dimensional. The implementation details of the PICProp-based methods are summarized in Table J.1, whereas 4,000 steps of standard PINN training are run for each BC for ES. For a fair comparison, 121 query points for PICProp are selected in the domain, see Figure J.1, and the computational time is approximately the same as of ES with 1,000 trials. Moreover, we include the ES result with 5,000 trials, which is considered to be exhaustive, as a reference.

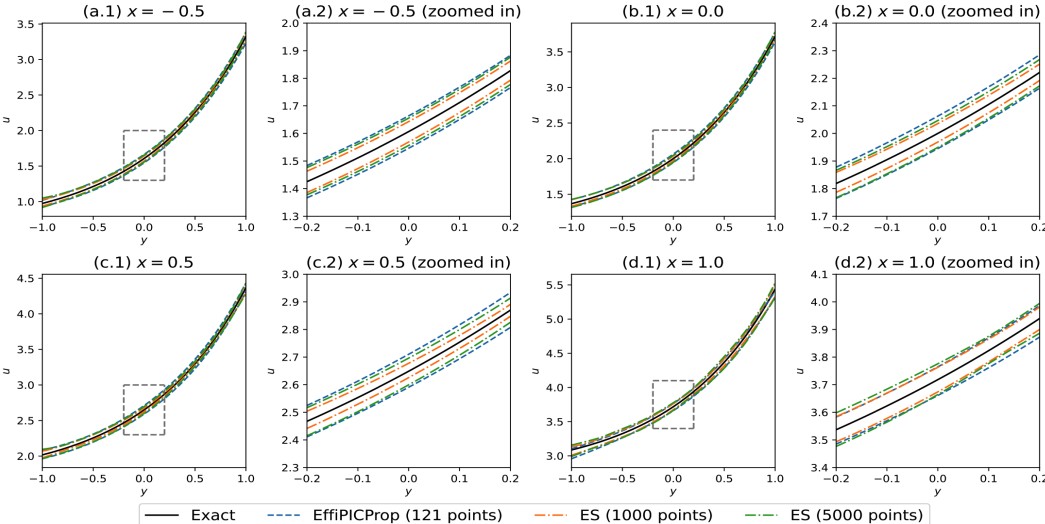

Figure 4: **2-D Poisson equation**: Slices of CI predictions for $x \in \{-0.5, 0.0, 0.5, 1.0\}$. EffiPICProp can identify the solution to the bi-level optimization problem more efficiently and accurately, as compared with the computationally expensive, search-based method ES.

Figure 4 plots the slices of CI predictions for $x \in \{-0.5, 0.0, 0.5, 1.0\}$, whereas Appendix K.2 includes slices along the $y$ axis and the heatmaps over the entire domain. For reference, the size

of the CIs obtained with EffiPICProp is approximately 0.1 across the domain, which is equal to the size of the propagated CI of Equation (12). The predicted CI size is also larger in most cases than that of the ES-based CIs. We conclude that 1,000 trials for ES are clearly not sufficient, as the intervals of ES with 5,000 trials are significantly wider; thus, a global optimum is not reached with 1,000 trials. In contrast, EffiPICProp converges to similar or wider intervals compared to ES with 5,000 trials. Overall, the optimization-based method EffiPICProp can identify the solution to the bi-level optimization problem more efficiently and accurately, as compared with the computationally expensive, search-based method ES.

### 5.3 One-dimensional, time-dependent Burgers equation

In this example, we study the Burgers equation given as

$$
\begin{aligned}
u_t + u_x - \nu u_{xx} = 0, \quad (x,t) \in [-1,1] \times [0,1], \quad \text{(PDE)}, \\
u(x, t = 0) = -\sin(\pi x), \quad u(x = -1, t) = u(x = 1, t) = 0, \quad \text{(BCs)},
\end{aligned}
\tag{13}
$$

where $\nu = \frac{0.01}{\pi}$ is the viscosity parameter. We consider clean data for the BCs and noisy data for the initial conditions (ICs). Specifically, for the ICs, we propagate the following fixed CI from $t = 0$ to the rest of the time domain $t \in (0, 1]$ and for all $x \in [-1, 1]$:

$$
u(x, t = 0) \in [-\sin(\pi x) - 0.2, -\sin(\pi x) + 0.2].
\tag{14}
$$

This problem serves to demonstrate the reliability of our method even in challenging cases with steep gradients or discontinuities of the PDE solution. For solving the problem using PINNs, we consider 200 boundary points, 256 initial points, and 10,000 samples in the interior domain. The 256 initial points are evenly spaced along $[-1, 1]$. The implementation details of the PICProp-based methods are summarized in Table J.1, whereas 2,500 and 5,000 trials are considered for ES. The 246 PICProp query points are visualized in Figure J.1 and the respective computational time is approximately equal to that of ES with 2,500 trials.

Slices of CI predictions for $t \in \{0.0, 0.25, 0.5, 0.75\}$ are plotted in Figure 5, whereas Appendix K.3 includes the respective meta-learning curves. We conclude that EffiPICProp, although computationally cheaper, in most cases provides wider, i.e., more conservative, CIs, as compared to ES with varying numbers of trials. It also succeeds in capturing the expected increase in uncertainty as the query points move away from the source of uncertainty, which corresponds to larger $t$ values. Specifically, the size of the CIs at $t = 0$ is approximately 0.4, which is equal to the size of the propagated CI in Equation (14), and becomes approximately 2 at $t = 0.75$ and near $x = 0$. On the other hand, since we consider 256 random initial points, the search space of ES is 256-dimensional. It requires a prohibitively large number of trials to identify the global optima of Equation (4). Finally, note that the results of EffiPICProp in Figure 5 correspond to $\lambda = 0$ in Equation (7), because we found that the additional information provided by the second loss term in Section 4.2 adversely affects generalization. This is partly due to the large number of query points in this problem.

## 6 Conclusion

Physics-informed machine learning is transforming the computational science field in an unprecedented manner, by solving ill-posed problems, involving noisy and multi-fidelity data as well as missing functional terms, which could not be tackled before. However, uncertainty quantification is necessary for deploying such methods in critical applications. In this work, we considered a novel problem pertaining to propagating randomness, in the form of CIs, from data locations (at the boundary but can easily be extended to inner or collocation data points) to the entire domain, with probabilistic guarantees. We proposed PICProp, a method based on bi-level optimization, and its efficient version, with a theorem guaranteeing the validity of the obtained CIs (Algorithms 1-2 and Theorem 4.1). To the best of our knowledge, this is the first work on joint CI estimation with theoretical guarantees in the context of physics-informed learning, while our method can be applied in diverse practical scenarios without imposing any distributional assumptions.

In the computational experiments of Section 5, involving various partial differential equations, we demonstrated the effectiveness and efficiency of our proposed methods. Specifically, the predicted CIs are wider, i.e. more conservative, in most cases, as compared with the more computationally demanding exhaustive search for solving the bi-level optimization. Although not encountered in the

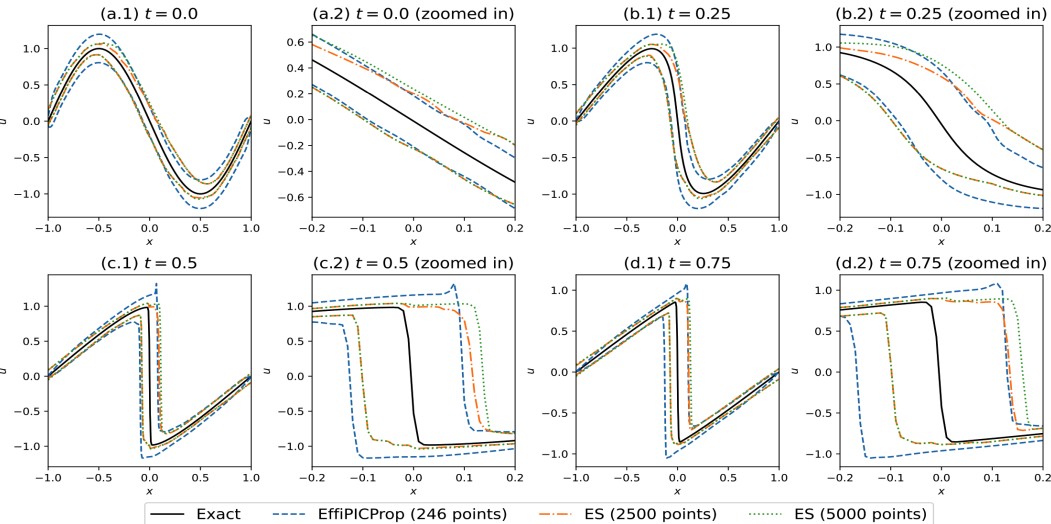

Figure 5: **Burgers equation**: Slices of CI predictions for $x \in \{0.0, 0.25, 0.5, 0.75\}$. EffiPICProp, although computationally cheaper, in most cases provides wider, i.e., more conservative CIs, as compared to ES with varying numbers of trials. To interpret the cost of ES, note that its search space is 256-dimensional and requires a prohibitively large number of trials to identify the global optima of Equation (4).

experiments, a potential limitation of our method is that the obtained CIs can be loose in some cases, even when propagating a tight CI from the boundaries, due to the max/min operations in the bi-level optimization of Equation (3). Because of the theoretical challenges involved, we plan to study this limitation and perform pertinent experiments in a future work.

## 7 Acknowledgements

This material is based upon work supported by the Google Cloud Research Credit program with the award (6NW8-CF7K-3AG4-1WH1). The computational work for this article was partially performed on resources of the National Supercomputing Centre, Singapore (https://www.nscc.sg).

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

## A  Indicative applications

Our proposed method, delineated in Algorithms 1 and 2 of Appendix H, provides a way to quantify how the data uncertainty affects the obtained solution. In addition, if the unseen clean data lies within the constructed/given data bounds, the ground-truth solution lies within our predicted CIs. Some indicative applications include:

1. In additive manufacturing problems, understanding, through our computed CIs, how the model responds to different material properties arising from the manufacturing process can help in model correction and thus quality control (see [31]).

2. Further, in material science, cracks affect the elasticity constants which together with geometric properties affect deformations (see [3]). The range of these deformations can be identified with our method. Accurately predicting the upper bound of such deformations can help designers to adjust cross-sectional properties of structural elements to better account for potential cracks.

3. More generally, our model can be extended to account for PDE model uncertainty. For example, PINNs have been used for short-term predictions of infected cases and deaths due to COVID-19 (see [6]). In such problems, our approach can be used to provide bounds of uncertainty for the predicted variables of interest. Accurately predicting the upper bound of infected cases can help the states better prepare for short-term hospitalization needs.

## B  Additional details regarding standard UQ

Standard UQ considers that the data is generated from a distribution $p(u|x, \theta = \theta^*)$, where $\theta^*$ is the NN parameters that are assumed to represent exactly the PDE solution $u(x)$. Given a dataset $z$, standard UQ approximates the solution $u_{\theta^*}(x)$ with the distribution

$$p(u|x; z) = \int p(u|x, \theta)p(\theta|z)d\theta, \tag{15}$$

where $p(\theta|z) \propto p(\theta)p(z|\theta)$ according to Bayes' theorem and $p(z|\theta)$ is the data likelihood. However, as a closed-form $p(\theta|z)$ is typically intractable, UQ draws approximate samples from $p(\theta|z)$, denoted as $\{\hat{\theta}_j\}_{j=1}^{N_\theta}$. Next, it constructs $\hat{p}(u|x; z)$ for each $x$ based on the collected samples to approximate $p(u|x; z)$. Typically, a Gaussian distribution is fitted to the PINN solutions $\{u_{\hat{\theta}_j}(x)\}_{j=1}^{N_\theta}$ and its corresponding statistical intervals are used in lieu of CIs for the PDE solution. Loosely speaking, statistical intervals of $\{\hat{\theta}_j\}_{j=1}^{N_\theta}$ can be viewed as credible intervals from Bayesian statistics, and intervals of $\{u_{\hat{\theta}_j}(x)\}_{j=1}^{N_\theta}$ as the corresponding prediction intervals.

Alternatively, one could use a concentration inequality to construct directly a prediction interval using the samples $\{u_{\hat{\theta}_j}(x)\}_{j=1}^{N_\theta}$. Nevertheless, such intervals still correspond to the Bayesian estimate $\mathbb{E}_{\theta \sim p(\theta|\mathcal{D})}u_\theta$, rather than to the exact solution $u(x)$, unless further assumptions are made. Even if the assumption that the Bayes estimate corresponds to $\theta^*$ is made, i.e., $\theta^* = \mathbb{E}_{\theta|\mathcal{D}}\theta$, the typically considered $\mathbb{E}_{\theta|\mathcal{D}}u_\theta(x)$ is not guaranteed to equal $u_{\mathbb{E}_{\theta|\mathcal{D}}\theta}(x)$ as $u_\theta$ can be arbitrary nonlinear (see also Appendix C).

Standard UQ approaches have several other limitations. For example, strong assumptions regarding the noise distribution and data likelihood are often required, and depending on the problem it is notoriously difficult to tune Bayesian NNs [5]. Furthermore, it is not possible to obtain/sample $p(u|x, z)$ exactly in practical problems and thus, additional approximations are required, typically referred to as *approximate posterior inference*. In addition, UQ approaches for PINNs are not supported by theoretical guarantees regarding the quality of uncertainty propagation due to the physics-informed constraints [11]. Finally, for the case of multiple available datasets, see Appendices C and E for the limitations of Monte Carlo-based approaches.

## C  Additional details regarding Monte Carlo Simulation

Assume that we have $m$ i.i.d. datasets denoted as $\{z^{(j)}\}_{j=1}^m$ and a 0-proper data-driven PDE solver (see Definition 3.3). It is natural to consider solving the PDE with each dataset in a Monte Carlo manner and then using a concentration inequality to construct CIs. However, a CI for the exact solution $u(x)$ cannot be identified without further assumptions. The reason is that we only have access to $\mathbb{E}[u(x; Z)]$, rather than to $u(x)$ given as $u(x) = u(x; \bar{Z}) = u(x; \mathbb{E}[Z])$, considering data-driven solutions from a 0-proper solver. Clearly, the difference between the two is in the form of $\mathbb{E}[f(X)] \neq f(\mathbb{E}[X])$, where $f$ in our case is the solver. In Figure C.1 we provide an illustration of this point as applied to linear/convex/concave functions, as s straightforward consequence of Jensen's inequality.

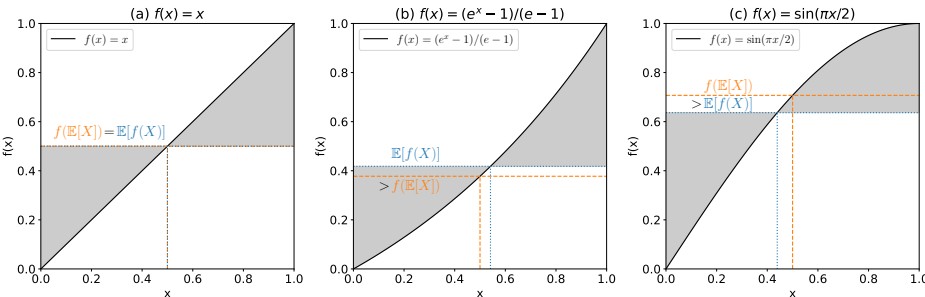

Figure C.1: Difference between $\mathbb{E}[f(X)]$ and $f(\mathbb{E}[X])$. (a) Linear function $f(x) = x$, $\mathbb{E}[f(X)] = f(\mathbb{E}[X])$; (b) Convex function $f(x) = (e^x - 1)/(e - 1)$, $\mathbb{E}[f(X)] > f(\mathbb{E}[X])$; (c) Concave function $f(x) = \sin(\pi x/2)$, $\mathbb{E}[f(X)] < f(\mathbb{E}[X])$.

## D   Confidence Interval Construction with Data

Consider a dataset $\{z^{(j)}\}_{j=1}^m$, which contains $m$ independent samples of an $n$-dimensional random vector $Z$. Based on the data $\{z^{(j)}\}_{j=1}^m$, we construct a CI for $Z$, denoted as $\tilde{\mathcal{Z}}_p$, where $p$ is the confidence level. Equivalently, $Pr(\bar{z} \in \tilde{\mathcal{Z}}_p) \geq p$.

In this regard, we consider the following problem scenarios:

1. The random vector $Z$ follows a multivariate Gaussian distribution with known covariance matrix $\Sigma$ of rank $n$. Given one observation $z$, i.e., $m = 1$, the chi-squared statistic $\chi^2 = (z - \mu)^T \Sigma^{-1}(z - \mu)$ follows a chi-squared distribution with $n$ degrees of freedom. The corresponding $p$-CI is $\tilde{\mathcal{Z}}_p = \{z' | \, ||z - z'||_{\Sigma^{-1}} \leq k_{n,p}^{\chi^2}\}$, where $||z - z'||_{\Sigma^{-1}} = \sqrt{(z - z')^T \Sigma^{-1}(z - z')}$ and $k_{n,p}^{\chi^2}$ is the squared root of the upper $(1 - p)/2$ critical value of the chi-squared distribution with $n$ degrees of freedom; see Figure 1 for an example corresponding to the case of $n = 2$.

2. The random vector $Z$ follows a multivariate Gaussian distribution with unknown covariance matrix $\Sigma$ and $m$ observations are available. The sample mean and covariance are estimated as $\bar{z} = \frac{1}{m}\sum_{i=1}^m z_i$ and $\hat{\Sigma} = \frac{1}{m-1}\sum_{i=1}^m (z_i - \bar{z})(z_i - \bar{z})^T$, respectively. The Hotelling's $t$-squared statistic $t^2 = N(\bar{z} - \mu)^T \hat{\Sigma}^{-1}(\bar{z} - \mu)$ follows a Hotelling's $T$-squared distribution with parameters $n$, $m - 1$. The corresponding $p$-CI is $\tilde{\mathcal{Z}}_p = \{z' | \, ||\bar{z} - z'||_{\hat{\Sigma}^{-1}} \leq m^{-1/2}k_{n,m-1,p}^{T^2}\}$, where $||\bar{z} - z'||_{\hat{\Sigma}^{-1}} = \sqrt{(\bar{z} - z')^T \hat{\Sigma}^{-1}(\bar{z} - z')}$ and $k_{n,m-1,p}^{T^2}$ denotes the squared root of the upper $(1 - p)/2$ critical value of the aforementioned Hotelling's $T$-squared distribution; see Figure 1 for an example corresponding to the case of $n = 2$ and $m = 5$.

3. The components of the random vector $Z$ follow an unknown distribution, and $m$ independent observations are available. For this case, a concentration inequality can be used for each component, i.e., for each data location, and the obtained bounds can be combined using a union of bounds for obtaining a $p$-CI (see also Appendix E).

   Note that many popular concentration inequalities require specific assumptions for the unknown distribution besides the fact that independent samples are available. For example, Hoeffding's inequality and its variants require the random variable to be bounded. [2] extends Hoeffding's inequality to unbounded random variables, but it is limited to Pareto-type tails. Other concentration inequalities without heavy assumptions for the distribution, such as Chebyshev's inequality and Chernoff's inequality, require various true population statistics (e.g., the population mean and variance) for computing the bounds, which are not available in the cases we consider. In this regard, [27] provides a Chebyshev-type bound with estimated mean and variance. [29] further extends the bound of [27] to the multivariate case.

4. The CI $\tilde{\mathcal{Z}}_p$ can alternatively be given by domain experts.

## E   Joint Confidence Intervals for PDE Solutions

In this section, we provide the necessary definitions, lemmas, and remarks that are required for introducing and addressing the CI propagation problem in Section 4.1. That is, how to propagate, with probabilistic guarantees, a CI constructed at the locations of the available data to arbitrary sets of query points of the domain.

To this aim, we first define the marginal CI for a single query point $x_q$.

**Definition E.1** (Marginal CI). A pair of random variables $(L, U) \in \mathbb{R}^2$ is a $p$-CI for $u(x_q)$ on the single query point $x_q \in \Omega$, if the random event $\mathcal{E} := \{L < u(x_q) < U\}$ holds with probability at least $p$.

Intuitively, considering repeating experiments of drawing random datasets $\{z^{(j)}\}_{1:m}$, the proportion of the $(L, U)$ CIs, constructed according to Definition E.1, that contain $u(x_q)$ is at least $p$. Next, we define the joint CI for multiple query points.

**Definition E.2** (Joint CI). A pair of functions $(L, U) \in \mathbb{R}^{\mathcal{X}_q} \times \mathbb{R}^{\mathcal{X}_q}$ is a $p$-CI for $u$ on query set $\mathcal{X}_q \subseteq \Omega$, if the random event $\mathcal{E} := \cap_{x_q \in \mathcal{X}_q} \{L(x_q) < u(x_q) < U(x_q)\}$ holds with probability at least $p$.

For simplicity, in the rest of the paper we refer to the $p$-CI for $u$ on the whole domain $\Omega$ as $p$-*CI for $u$*, and to the $p$-CI for $u$ on $\mathcal{X}_q \subset \Omega$ as *query set $p$-CI*.

*Remark* E.3. Let $(L, U)$ be a $p$-CI for $u$. For any $\mathcal{X}_q \subseteq \Omega$, $(L, U)$ is a $p$-CI for $u$ on $\mathcal{X}_q$, i.e., the random event $\mathcal{E}(\mathcal{X}_q, Z) := \cap_{x_q \in \mathcal{X}_q} \{L(x_q) < u(x_q) < U(x_q)\}$ holds with probability at least $p$.

Remark E.3 indicates that a $p$-CI for $u$ on an arbitrary query set can be derived from a $p$-CI for $u$ by slicing. The query set $\mathcal{X}_q$ can be either discrete or continuous including multiple or even infinite queries. Note that the $p$-CIs of Definition E.2 and Remark E.3 hold with probability at least $p$; i.e., there is no statement about their tightness. In this regard, the next remark shows that slicing using larger sets $\mathcal{X}_q$ yields tighter $p$-CIs for $u$ on $\mathcal{X}_q$. Nevertheless, the maximum tightness that can be achieved, for the $p$-CI for $u$ on $\mathcal{X}_q$, is determined by the tightness of the $p$-CI for $u$ on the whole domain $\mathcal{X}_q$. This is corroborated by the following remark.

*Remark* E.4. Let $(L, U) \in \mathbb{R}^\Omega \times \mathbb{R}^\Omega$ be a $p$-CI for $u$. For query set $\mathcal{X}_q^1 \subseteq \mathcal{X}_q^2 \subseteq \Omega$, the probabilities of random events $\mathcal{E}_1 := \cap_{x_q \in \mathcal{X}_q^1} \{L(x_q) < u(x_q) < U(x_q)\}$, $\mathcal{E}_2 := \cap_{x_q \in \mathcal{X}_q^2} \{L(x_q) < u(x_q) < U(x_q)\}$ satisfy

$$\Pr(\mathcal{E}_1 \geq \Pr(\mathcal{E}_2). \tag{16}$$

Next, we define the union of single-query $p$-CIs. It is often used in practice as a proxy to the query set $p$-CI and can be used also within the context of the present study for addressing problem scenarios 3 and 4, as described in Appendix D.

**Definition E.5** (Union of CIs). A pair of functions $(L, U) \in \mathbb{R}^{\mathcal{X}_q} \times \mathbb{R}^{\mathcal{X}_q}$ constructed based on $\{z^{(j)}\}_{1:m}$ is a union of $p$-CIs for $u$ on query set $\mathcal{X}_q \subseteq \Omega$ if for each $x_q \in \mathcal{X}_q$, $(L(x_q), U(x_q))$ is a $p$-CI for $u(x_q)$, i.e., each random event $\mathcal{E} := \{L(x_q) < u(x_q) < U(x_q)\}$ holds with at least probability $p$.

The next lemma provides insight into the connection between the marginal $p$-CI on $\mathcal{X}_q$ and the union of $p$-CIs.

**Lemma E.6.** *For $u$ and $\mathcal{X}_q$, a $p$-CI is a union of $p$-CIs, whereas a union of $p$-CIs is not necessarily a $p$-CI.*

That is, $u(x_q), \forall x_q \in \mathcal{X}_q$, belongs to the $p$-CI constructed by slicing of the $p$-CI for $u$ with probability at least $p$. On the other hand, the probability of the event $\cap_{x_q \in \mathcal{X}_q} \{L(x_q) < u(x_q) < U(x_q)\}$, where $(L, U)$ are constructed based on a union of $p$-CIs, may be less than $p$. This is supported by the following two remarks as well as the computational experiment of Appendix F.

*Remark* E.7. The probability of the random event $\cap_{x_q \in \mathcal{X}_q} \{L(x_q) < u(x_q) < U(x_q)\}$, where $(L, U)$ are constructed based on a union of $p$-CIs, is upper-bounded by the minimum probability of the events $\{L(x_q) < u(x_q) < U(x_q)\}, \forall x_q \in \mathcal{X}_q$. That is,

$$\Pr(\cap_{x_q \in \mathcal{X}_q} \{L(x_q) < u(x_q) < U(x_q)\}) \\ \leq \min_{x_q \in \mathcal{X}_q} \Pr(\{L(x_q) < u(x_q) < U(x_q)\}). \tag{17}$$

*Remark* E.8. If one of the $p$-CIs is tight, i.e. $\exists x_q' \in \mathcal{X}_q$ such that $\Pr(\{L(x_q') < u(x_q') < U(x_q')\}) = p$, the union of these $p$-CIs is a $p$-CI for $u$ if and only if it holds $\forall x_q \in \mathcal{X}_q \setminus \{x_q'\}$ that

$$\Pr(\{L(x_q) < u(x_q) < U(x_q) | L(x_q') < u(x_q') < U(x_q')\}) = 1. \tag{18}$$

Finally, in order to obtain a guarantee that a union of CIs is a $p$-CI, we can equate a lower bound of the probability of the former with $p$. This lower bound can be obtained by disregarding correlations among the different locations, as formalized in the following lemma.

**Lemma E.9.** *For $u$ and finite query set $\mathcal{X}_q$, a union of $(1 - (1-p)/|\mathcal{X}_q|)$-CIs is a $p$-CI for $u$ as*

$$\Pr(\cap_{x_q \in \mathcal{X}_q} \{L(x_q) \leq u(x_q) \leq U(x_q)\})$$
$$= 1 - \Pr(\cup_{x \in \mathcal{X}_q} \{L(x) \leq f(x) \leq U(x)\}^C)$$
$$\geq 1 - \sum_{x \in \mathcal{X}_q} (1 - \Pr(\{L(x) \leq u(x) \leq U(x)\})) \tag{19}$$
$$\geq 1 - |\mathcal{X}_q|(1 - (1 - \frac{1-p}{|\mathcal{X}_q|})) = p.$$

However, as the size of the query set $\mathcal{X}_q$ increases, the required confidence level of each interval must approach 1, which leads to loose union bounds. Finally, another approach for obtaining a $p$-CI for multiple queries is to use multivariate concentration inequalities [8]. Nevertheless, this approach is limited to a finite number of queries.

# F  Toy Linear System

In this section, we highlight the differences between joint CIs and union of CIs, as described in Appendix E, with a simple example. Consider a true function $u$ and a model given as $u_\theta(x) = (1-x)\theta_0 + x\theta_1, x \in [0,1]$, with $\theta = (\theta_0, \theta_1) \in \mathbb{R}^2$. We assume that there exists $\theta^*$ such that $u_{\theta^*} = u$, and that we are given measurements $Z = (Z_0, Z_1) \sim \mathcal{N}(\theta^*, I)$. Thus, for one observation $Z$, we simply estimate $\hat{\theta} = Z$. As a result, the estimator for $u$ is given as

$$u_{\hat{\theta}} = (1-x)Z_0 + xZ_1 \sim \mathcal{N}((1-x)\theta_0^* + x\theta_1^*, 2x^2 - 2x + 1). \tag{20}$$

Next, define as $k$-interval the lower and upper intervals $(L_k, U_k)$ obtained by subtracting and adding $k$ standard deviations from the estimated mean of $u$, respectively, i.e.,

$$
\begin{aligned}
L_k(x) &:= (1-x)Z_0 + xZ_1 - k\sqrt{2x^2 - 2x + 1}, \\
U_k(x) &:= (1-x)Z_0 + xZ_1 + k\sqrt{2x^2 - 2x + 1}.
\end{aligned}
\tag{21}
$$

*Remark* F.1. We define $k_p^{\mathcal{N}}$-interval as the interval constructed as a union of one-query $p$-CIs for $u_\theta(x)$ and for different values of $x \in \Omega$, similarly to Definition E.5. The value $k_p^{\mathcal{N}}$ is the upper $(1-p)/2$ critical value of the standard normal distribution.

*Proof*: For $x \in \Omega$,

$$
\begin{aligned}
&\Pr(\{L_{k_p^{\mathcal{N}}} \le u(x) \le U_{k_p^{\mathcal{N}}}\}) \\
=&\Pr(\{(1-x)Z_0 + xZ_1 - k_p^{\mathcal{N}}\sqrt{2x^2 - 2x + 1} < f(x) < (1-x)Z_0 + xZ_1 + k_p^{\mathcal{N}}\sqrt{2x^2 - 2x + 1}\} \\
=&\Pr(\{|u_{\hat{\theta}}(x) - \mathbb{E}[u_{\hat{\theta}}(x)]| < k_p^{\mathcal{N}}\mathrm{std}(u_{\hat{\theta}}(x))\} \\
=&p.
\end{aligned}
\tag{22}
$$

*Remark* F.2. We define $k_p^{\chi^2}$-interval as the $p$-CI for $u$, constructed as $(L_{k_p^{\chi^2}}(x), U_{k_p^{\chi^2}}(x))$ for each $x \in \Omega$, where

$$
\begin{aligned}
L_{k_p^{\chi^2}}(x) &= \underset{||z - Z||_2 \le k_p^{\chi^2}}{\arg\min} (1-x)z_0 + xz_1, \\
U_{k_p^{\chi^2}}(x) &= \underset{||z - Z||_2 \le k_p^{\chi^2}}{\arg\max} (1-x)z_0 + xz_1.
\end{aligned}
\tag{23}
$$

The value $k_p^{\chi^2}$ is the squared root of the upper $(1-p)/2$ critical value of the chi-squared distribution with 2 degrees of freedom.

*Proof*: According to Theorem 4.1,

$$
\begin{aligned}
&\Pr(\{L_{k_p^{\chi^2}}(x) < u(x) < U_{k_p^{\chi^2}(x)}, \forall x \in \Omega\}) \\
\ge&\Pr(\{||\theta^* - Z||_2 \le k_p^{\chi^2}\}) \\
=&p.
\end{aligned}
\tag{24}
$$

The left side of Figure F.1 provides an indicative example of a $k_p^{\mathcal{N}}$-interval and a $k_p^{\chi^2}$-interval, with $p = 95\%$ and $\theta^* = (0,0)$. The results are obtained using only one sample $Z = (Z_0, Z_1)$, which can be visualized by considering that $u_\theta(0) = Z_0$ and $u_\theta(1) = Z_1$. Further, the empirical confidence level of an interval $(L, U)$ can be estimated as $\bar{p} := \frac{1}{J}\sum_{j=1}^J \mathbf{1}(\cap_{k=0,1,...,\lfloor 1/\epsilon \rfloor}\{L_{Z_j}(k\epsilon) < u(k\epsilon) < U_{Z_j}(k\epsilon)\})$ with $\epsilon > 0$. That is, we calculate the proportion of times that such a construction of the interval contains the true values on $\{0, \epsilon, ..., \lfloor 1/\epsilon \rfloor \epsilon\}$. According to the law of large numbers, the empirical confidence level converges with increasing $J$, in probability, to the true confidence level $p$. The right side of Figure F.1 shows that the empirical confidence level curve of the $k_p^{\chi^2}$-interval (or the $k_p^{\mathcal{N}}$-interval) converges to a value above (or below) 95%. This result implies that the $k_p^{\chi^2}$-interval is a valid 95%-CI for $u$ while the $k_p^{\mathcal{N}}$-interval is not.

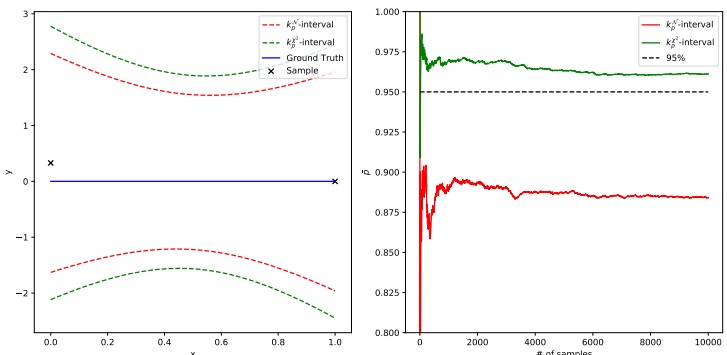

Figure F.1: Left: An indicative example of a $k_p^{\mathcal{N}}$-interval and a $k_p^{\chi^2}$-interval, with $p = 95\%$ and $\theta^* = (0,0)$. The results are obtained using only one sample $Z = (Z_0, Z_1)$, which can be visualized by considering that $u_\theta(0) = Z_0$ and $u_\theta(1) = Z_1$. Right: Empirical confidence level on $\Omega$ versus the number of samples $J$.

## G  Proofs

### G.1  Proof of Theorem 4.1

If $\bar{z} \in \tilde{\mathcal{Z}}_p$, then $L(x) = \min_{z \in \tilde{\mathcal{Z}}_p} u(x; \bar{z}) \leq u_{\mathcal{A}}(x; \bar{z}) \leq \max_{z \in \tilde{\mathcal{Z}}_p} u(x; \bar{z}) = U(x)$, for all $x \in \Omega$. Further, according to the definition of the proper solver, we have $u_{\mathcal{A}}(x; \bar{z}) - \eta \leq u(x) \leq u_{\mathcal{A}}(x; \bar{z}) + \eta$, for all $x \in \Omega$, and thus $L(x) - \eta \leq u(x) \leq U(x) + \eta$, for all $x \in \Omega$. Finally, we can conclude that $Pr\{L(x) - \eta \leq u(x) \leq U(x) + \eta, \forall x \in \Omega\} \geq Pr\{\bar{z} \in \tilde{\mathcal{Z}}_p\} \geq p$.

## H  Algorithms

The developed methods are summarized in Algorithms 1 and 2.

---

**Algorithm 1** PICProp: Physics-Informed Confidence Propagation (Gradient-based)

---

**Input:** $\tilde{\mathcal{Z}}_p$($p$-CI for $\tilde{z}$), query point $x$, inner-steps schedule $\{N_k\}_{k=1}^{K_o}$, learning rates $\alpha, \beta$, outer steps $K_o$.
Initialize $z_0 \in \tilde{\mathcal{Z}}_p$, $\theta_0$.
**for** $k = 1, ..., K_o$ **do**
  **for** $n = 1, ..., N_k$ **do**
    $\theta_n = \theta_{n-1} - \alpha \nabla_\theta \mathcal{L}_{pinn}(\theta_{n-1}, z_{k-1})$
  **end for**
  $\hat{\theta} = \theta_0 = \theta_{N_k}$
  **if** lower bound is sought **then**
    $z_k = z_{k-1} - \beta \nabla_z u_{\hat{\theta}(z_{k-1})}(x)$
  **else**
    $z_k = z_{k-1} + \beta \nabla_z u_{\hat{\theta}(z_{k-1})}(x)$
  **end if**
**end for**
**return** $u_{\hat{\theta}(z_{K_o})}(x)$

---

## I  Bi-level Optimization

A bi-level optimization problem can be formulated as

$$\min_\phi \ \mathcal{J}(\theta^*(\phi), \phi) \quad s.t. \ \theta^*(\phi) = \arg\min_\theta \mathcal{L}(\theta, \phi), \tag{25}$$

**Algorithm 2** EffiPICProp: Efficient Physics-Informed Confidence Propagation

---

**Input:** $\eta, p(x_q), \lambda$.
Randomly sample $x_q^{1:K} \sim p(x_q)$.
Obtain $\hat{\theta}(z^L(x_q^k)), \hat{\theta}(z^U(x_q^k))$ with Algorithm 1 for $k = 1, ..., K$.
Initialize $\psi$.
**while** not done **do**
  $\psi \leftarrow \psi - \eta \nabla \mathcal{L}(x_q^{1:K}; \psi)$ according to Equation (7).
**end while**
**return** $u_\psi$

---

where $\theta, \phi$ are the parameters and the hyperparameters, whereas $\mathcal{L}$ and $\mathcal{J}$ are the lower-level objective and the upper-level objective. Many problems like meta-learning, hyperparameter optimization, data poisoning, and auxiliary learning can be naturally formulated in a bi-level optimization way.

The hardest part of bi-level optimization is that $\theta^*(\phi)$ is not available in a closed-form in many scenarios, particularly with an empirical lower-level objective that requires multiple steps of gradient descent. In such cases, the explicit gradient of high-level objective with respect to hyperparameter is intractable.

According to the chain rule,

$$\nabla_\phi \mathcal{J}(\theta^*(\phi), \phi) = \nabla_\theta \mathcal{J} \nabla_\phi \theta^*(\phi) + \nabla_\phi \mathcal{J}. \tag{26}$$

The most challenging part is the approximation of the implicit gradient $\nabla_\phi \theta^*(\phi)$, for which many advanced methods have been proposed and studied in recent years [15]. In this paper, three methods referred to as *Reverse*, *approximate implicit differentiation with Neumann Series* (AID-NS), and *approximate implicit differentiation with conjugate gradient* (AID-CG) are evaluated on the confidence interval identification tasks.

### I.1 Reverse

**Reverse** is an iterative differentiation-based method to approximate the hypergradient by unrolling the inner optimization $\theta_{1:K}$ trajectory in a reverse way. The implicit gradient can be approximated by

$$\nabla_\phi \theta^*(\phi) \approx \nabla_\phi \theta_K(\phi) \approx \sum_{k=1}^{K-1} \left( \prod_{i=k+1}^{K-1} \frac{\partial \tilde{\theta}_{i+1}}{\partial \theta_i} \right) \frac{\partial \tilde{\theta}_{k+1}}{\phi}, \tag{27}$$

where $\tilde{\theta}_{k+1} = \theta_k - \alpha \nabla_\theta l(\theta_k, \phi)$ and $\alpha$ is a pre-defined inner-learning rate.

### I.2 Approximate Implicit Differentiation (AID)

Another family of methods to approximate the hypergradient is Implicit Differentiation, where the implicit function theorem (IFT) is used to evaluate $\nabla_\phi \theta^*(\phi)$. If $\mathcal{L}$ is second-order derivable and strong convex, we have

$$\nabla_\phi \theta^*(\phi) = -(\nabla_\theta^2 \mathcal{L})^{-1} \nabla_\phi \nabla_\theta \mathcal{L}. \tag{28}$$

Then we can approximate the gradients of $\phi$ using

$$\nabla_\phi \mathcal{J}(\theta^*(\phi), \phi) = -\nabla_\theta \mathcal{J} \cdot (\nabla_\theta^2 \mathcal{L})^{-1} \nabla_\phi \nabla_\theta \mathcal{L} + \nabla_\phi \mathcal{J}. \tag{29}$$

The most challenging part is the approximation of the inverse Hessian vector product (inv-hvp) $A^{-1}p$, where $A \in \mathbb{R}^{d \times d}$ is a given symmetric positive matrix and $p \in \mathbb{R}$ is a given vector.

AID-NS [19] is a method based on the Neumann Series to approximate the inv-hvp as

$$A^{-1}p = \lambda \sum_{k=0}^{\infty} (I - \lambda A)^k p \approx \lambda \sum_{k=0}^{K} (I - \lambda A)^k pm \tag{30}$$

where $\lambda$ is a hyperparameter that ensures convergence and $K$ is a hyperparameter for precision control.

AID-CG [12] approximates the hypergradient by solving the system of homogeneous linear equations $Ax = p$ with conjugate gradient.

# J Details regarding implementation, hyperparameters, query points, and computational times

## J.1 Implementation and hyperparameters

For each PDE, we use a multilayer perceptron (MLP) for the PINN model and another MLP of the same architecture for EffiPICProp. Query points are collected in a grid manner. For each query point, PICProp starts with standard PINN training with a randomly initialized virtual boundary condition. We refer to this as the *warmup* phase. Subsequently, PICProp alternates between performing a number of inner steps to update the PINN model and computing the hyper-gradient for a meta step to update the virtual boundary condition. The inner/outer learning rate is selected from $\{0.001, 0.01, 0.1\}$. The inner/outer optimizer is selected from *SGD* and *Adam*. The hyper-gradient method is selected from *Neumann Series (NS)*, *Conjugate Gradient (CG)*, *Reverse*; see more details in Appendix I. The numbers of warmup/inner/meta steps are tuned to balance the stability of bi-level training and the computational cost. For the pedagogical example, as six points are too few for train-validation splitting, we manually set $\lambda = 1$ for EffiPICProp. In the rest of the examples, we split 10% of the training data as the validation set to select the best $\lambda$ from $\{0.0, 0.25, 0.5, 0.75, 1.0\}$. The hyperparameters are summarized in Table J.1.

Table J.1: Summary of implementation details, and utilized hyperparameters and architectures in the experiments.

|                | Pedagogical | Poisson | Burgers |
|----------------|-------------|---------|---------|
| MLP arch.      | $32 \times 2$ | $20 \times 8$ | $20 \times 8$ |
| # BC points    | 2           | 40      | 256     |
| # query points | 6           | $11 \times 11$ | $41 \times 6$ |
| inner optim.   | Adam        | Adam    | Adam    |
| inner lr       | 0.001       | 0.001   | 0.001   |
| meta optim.    | SGD         | SGD     | SGD     |
| meta lr        | 0.01        | 0.001   | 0.001   |
| hypergrad      | NS          | Reverse | Reverse |
| # warmup steps | 2,000       | 4,000   | 20,000  |
| # inner steps  | 500         | 20      | 100     |
| # meta steps   | 50          | 200     | 500     |
| $\lambda$      | 1.0         | 1.0     | 0.0     |

## J.2 Query Points

The query points are summarized in Figure J.1.

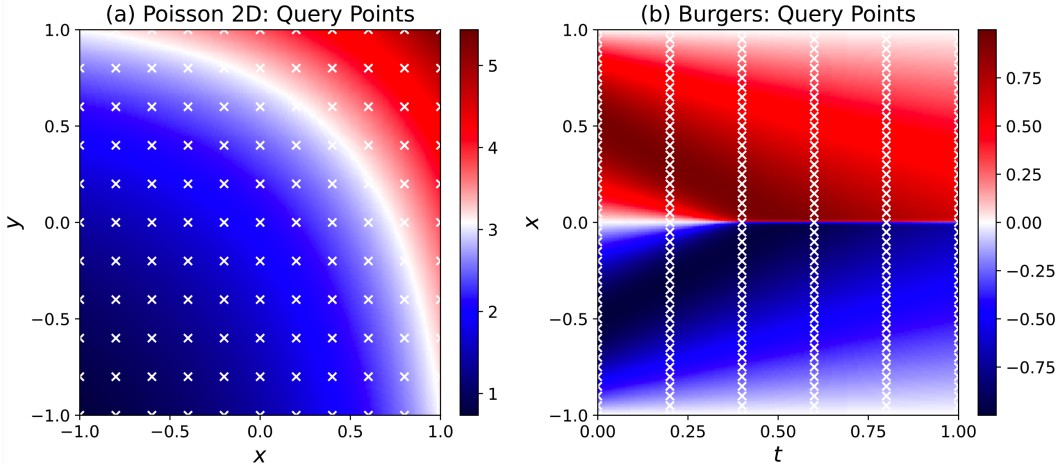

Figure J.1: Visualization of the query points on top of the exact solutions of Equations 11 and 13. (a) 2-D Poisson equation: 121 equally spaced points on a grid. (b) Burgers equation: 246 equally spaced points on a grid.

## J.3    Computational times

In our computational experiments, all methods were run serially and thus, the runtimes shown below are much larger than the required ones for using these methods in practice. For this reason, we have also included a "pseudo-parallel" runtime that shows the runtimes if parallel processing had been used instead. The runtimes for the examples of Section 5 are summarized in Table J.2.

Table J.2: Computational time for different experiments.

| Experiments | ES | PICProp | SimPICProp | EffiPICProp |
|---|---|---|---|---|
| Pedagogical example (CPU) | 42 hrs for 5,000 trials | 1.7 hrs for 41 queries | 16 mins (40s + 6 PICProp queries) | 18 mins (40s per lambda + 6 PICProp queries) |
| 2D Poisson equation (GPU) | 14 hrs for 1,000 trials 70 hrs for 5,000 trials | 400s per query | – | 14 hrs (180s per lambda + 121 PICProp queries), 10 mins for pseudo-parallel |
| 1D time-dependent Burgers equation (GPU) | 347 hrs for 2,500 trials 694 hrs for 5,000 trials | 5,000s per query | – | 14 hrs (2000s per lambda + 246 PICProp queries), 2 hrs for pseudo-parallel |

# K    Additional Experimental Results

## K.1    Pedagogical example

### K.1.1    Learning Curves

Figure K.1 provides the meta-learning curves corresponding to the CI predictions of Figure 2 (main text). Figure K.2 provides the results of all considered meta-learning methods and of ES on six query points. We conclude that the combination of *Neumann Series (NS)* and *SGD* stably converges to the reference result of ES.

### K.1.2    $\lambda$s

Figure K.3 provides comparison among PICProp and EffiPICProp with various $\lambda$s. Notice that EffiPICProp with $\lambda = 0$ is referred to as SimPICProp in Figure 3. The mean squared errors (MSE) between PICProp CI and EffiPICProp CI are summarized in Appendix K.1.4. We conclude that EffiPICProp CI with $\lambda = 1.0$ is the best approximation of PICProp CI.

### K.1.3    SimPICProp v.s. EffiPICProp

Figure K.4 and Figure K.5 provide the comparison between SimPICProp and EffiPICProp on $T^2$ and Hoeffding boundary CI, respectively.

### K.1.4    Validity of EffiPICProp CIs

Although it has been already presented in Figure 2 that PICProp and EffiPICProp predictions well match the ES results, we conduct the following experiment to further verify the validity of EffiPICProp CI predictions.

We independently sampled 500 boundary conditions and propagate the corresponding $\chi^2$ CIs using EffiPICProp to construct CIs for the entire domain. According to the definition of joint CIs (see Definition 3.1), an instance of CI is valid if it contains the solution $u(x)$ for any $x \in [-1, 1]$. In this regard, we uniformly sampled 101 data points from $[-1, 1]$ and regard an instance of CI as valid if the solution was enclosed within the interval for these data points. Further, we run vanilla PINNs with clean boundary data and use the max absolute error among these 101 data points as $\eta$. As a result, $486$ out of these 500 ($97.2\% > 95\%$) instances of CI are valid, which empirically indicates that EffiPICProp CI predicitons are valid and conservative.

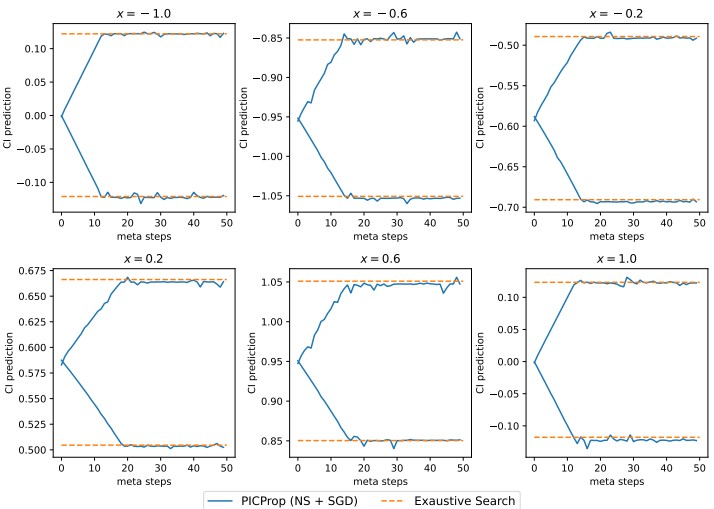

Figure K.1: **Pedagogical example:** The learning curve of PICProp with *Neumann Series (NS)* and *SGD* on six query points. Comparisons with exhaustive search results for solving the bi-level optimization problem for each query point are also provided.

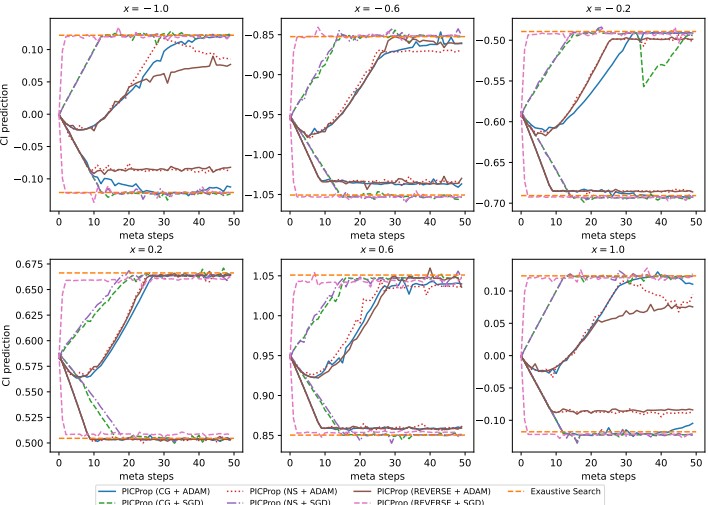

Figure K.2: **Pedagogical example:** Various combinations of hyper-gradient methods with meta-optimizers and comparisons with exhaustive search results for solving the bi-level optimization problem for each query point.

| $\lambda$ | 0.0 | 0.25 | 0.5 | 0.75 | 1.0 |
|---|---|---|---|---|---|
| MSE | 1.4E-3 | 2.2E-4 | 2.2E-4 | 1.7E-4 | **1.8E-5** |

Table K.1: Mean Squared Errors between PICProp CI and EffiPICProp CI. EffiPICProp CI with $\lambda = 1.0$ is the best approximation of PICProp CI.

## K.2   2-D Poisson equation

Figure K.6 includes slices along the $y$ axis and accompanies Figure 4 (main text), which includes slices along the $x$ axis. Heatmaps with CIs over the entire domain are presented in Figure K.7.

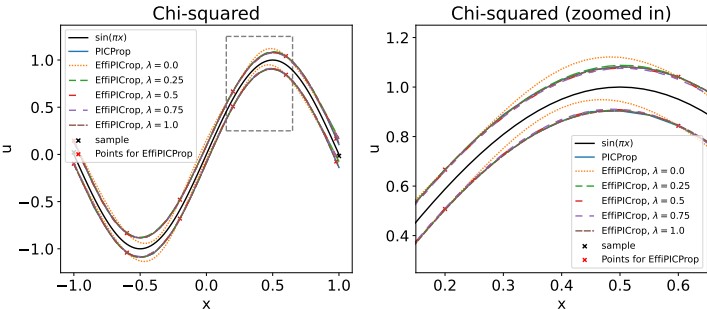

Figure K.3: **Pedagogical example**, $\chi^2$: Comparison among PICProp and EffiPICProp with various $\lambda$s. Notice that EffiPICProp with $\lambda = 0$ is referred to as SimPICProp in Figure 3
.

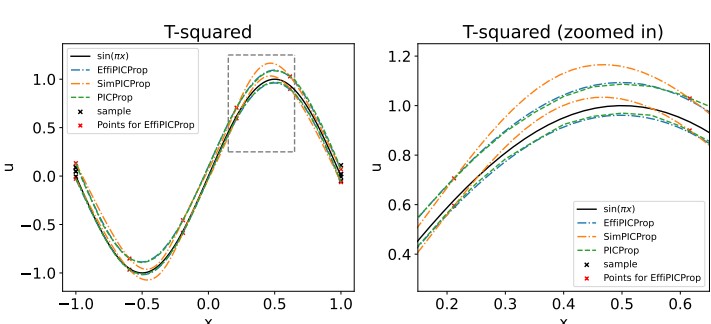

Figure K.4: **Pedagogical example**, $T^2$: Although the results of both SimPICProp and EffiPICProp match PICProp results on the query points, EffiPICProp generalizes better on unseen locations.

### K.3  Burgers equation

Figures K.8 & K.9 provide the meta-learning curves corresponding to the CI predictions of Figure 5 (main text).

## L  Comparison with Bayesian PINNs

In this section, we compare PICProp with Bayesian PINNs (BPINNs) [34] for the problem of Section 5.1. For BPINNs, the inference method for posterior estimation is Hamiltonian Monte Carlo (HMC) [4]. Although provided here for completeness, we believe that such a comparison can also be misleading because of the many differences between the two methods. The first difference pertains to the fact that PICProp does not consider parameter uncertainty. Further, the assumptions for these two methods are different: BPINN requires heavy assumptions about the data likelihood, for instance, while PICProp only requires the noise to be mean-zero. Furthermore, PICProp produces joint CIs for the whole domain, while the BPINN intervals are marginal. Finally, although supported by a sound Bayesian justification, BPINNs are not supported by a strict confidence guarantee for the produced intervals, in contrast to Theorem 4.1 that supports PICProp.

We summarize and contrast the two approaches in the following: BPINNs assume a NN parameter prior, and subsequently approximate and sample the parameter posterior to obtain samples from the solution posterior. This is done in order to fit a solution posterior distribution and subsequently compute its statistical intervals. In contrast, the problem of UQ is rethought ab initio in our paper. PICProp first considers either CIs given at the boundaries or data collected at the boundaries and estimates the corresponding CIs. Subsequently, the boundary CIs are propagated to the interior domain with mild assumptions and with the theoretical guarantee that the obtained CIs are indeed CIs according to the definition.

From the results summarized in L.1, we observe from Figure L.1 that the PICProp CIs are more conservative than the BPINN. From a design perspective, conservative uncertainty estimation is the preferred route for critical and risk-sensitive applications [7, 13], although it can be more costly. In contrast, most widely used UQ methods, such as deep ensembles [18] and Monte-Carlo dropout [28, 10], are known to be overconfident in practice [9, 7]. Finally, note that the BPINN results of Figure L.1 were obtained by using the software NeuralUQ developed in [38].

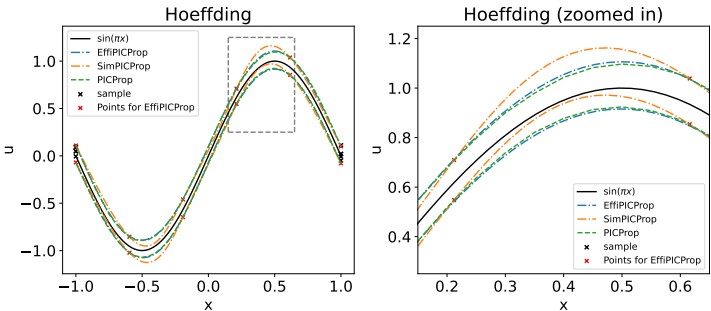

Figure K.5: **Pedagogical example**, Hoeffding: Although the results of both SimPICProp and EffiPICProp match PICProp results on the query points, EffiPICProp generalizes better on unseen locations.

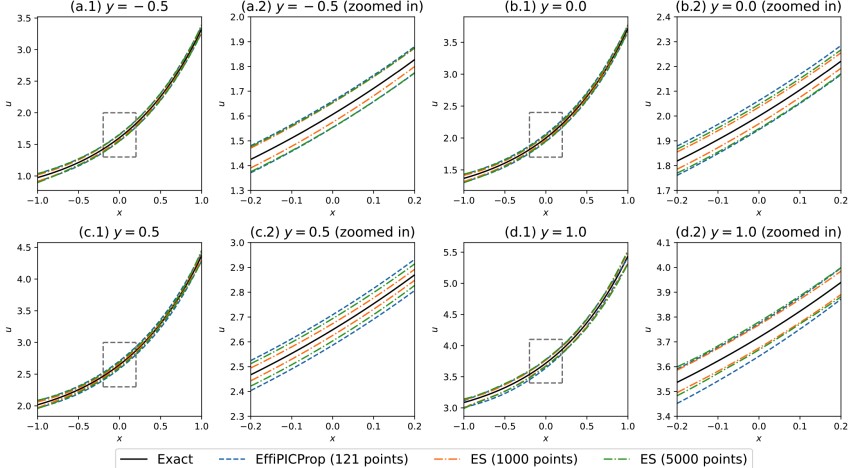

Figure K.6: **2-D Poisson equation**: Slices of CI predictions for $y \in \{-0.5, 0.0, 0.5, 1.0\}$.

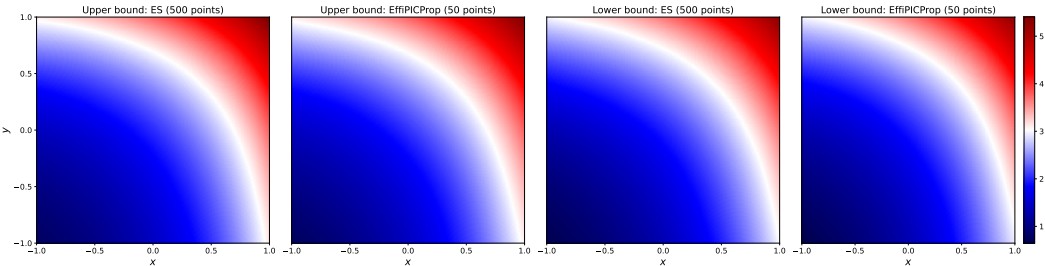

Figure K.7: **2-D Poisson equation**: Heatmaps of upper and lower CIs predicted by the exhaustive search (ES) and EffiPICProp.

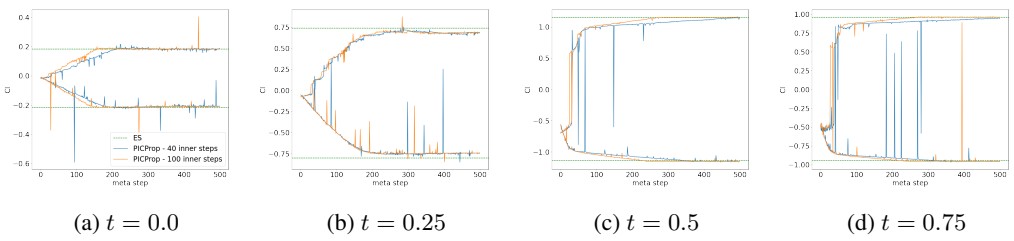

(a) $t = 0.0$      (b) $t = 0.25$      (c) $t = 0.5$      (d) $t = 0.75$

Figure K.8: **Burgers equation:** Meta-learning curves for $x = 0$ demonstrating the stability of our method.

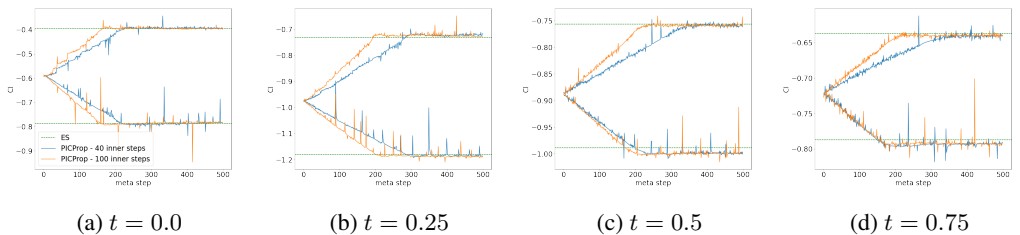

(a) $t = 0.0$    (b) $t = 0.25$    (c) $t = 0.5$    (d) $t = 0.75$

Figure K.9: **Burgers equation:** Meta-learning curves for $x = 0.2$ demonstrating the stability of our method.

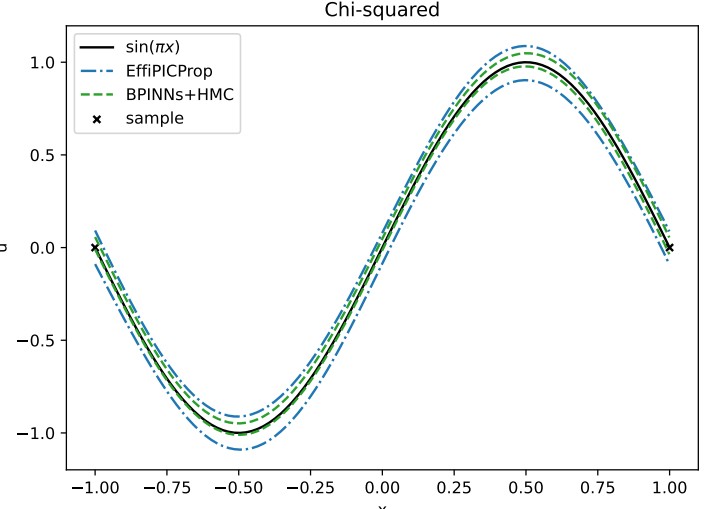

Figure L.1: **Pedagogical example**: The comparison between intervals produced by EffiPICProp and BPINNs [34]+HMC [4]. Noise on the boundary is Gaussian with known standard deviation $\sigma = 0.05$.

