# OpenReview forum: "PICProp: Physics-Informed Confidence Propagation for Uncertainty Quantification"
_NeurIPS.cc/2023/Conference — NeurIPS 2023 poster_

### Official Review · Reviewer_5z2U · 2023-06-15

**Soundness:** 3 good
**Presentation:** 2 fair
**Contribution:** 3 good
**Rating:** 5
**Confidence:** 3

**Summary:**

The paper presents methods to estimate confidence sequences for the solution of a differential equation given noisy boundary conditions. Various methods are proposed of different complexity and the confidence sequences are illustrated for the solution of three different problems.

**Strengths:**

The overall problem to determine the confidence sequence is laid out.  Under the given assumptions, the problem is well formed and an approximation of the exact solution via exhaustive search is clear.  The faster methods that incorporate meta-learning techniques and surrogate solutions for the lower and upper sequences are innovative and clever.

**Weaknesses:**

The main assumption of the paper is that the accuracy of the solution provided by the machine learning model is either perfect (\eta = 0) or \eta is known and fixed when the boundary conditions are given perfectly.  Even with clean boundary data, the accuracy of the model will depend of the number of internal and boundary training points, and these are not incorporated in the computation of the confidence sequences.

While it might be OK to consider the exhaustive search as the gold standard, the examples do not confirm whether or not the derived confidence sequences are indeed satisfying Definition 3.1. In light of the first comment, this should be validated.

Algorithm 1 and 2 are part of the main contribution of the paper.  However, they are tucked away in the supplementary appendix. As such, a reader if forced to go to the supplementary material to even understand the gist of the paper.

**Questions:**

In eqn. (5) it seems that the lower and upper boundary data points should be derived by finding the points that  minimized u_{\hat{\theta}}(x) at x = x_q^k rather than for all x.  Is that correct?

Four lines above eqn. (4), should z minimize (rather than maximize) the corresponding value to determine L(x)?

Third to last last line of section 4, SimPICProp is not defined in the main paper.  It may be better to state that SIMPICProp is the version of EffPICProp where \lambda = 0.

The paper use the term confidence interval for Def. 3.1.  Based on the work of Ramdas, I believe this is a confidence sequence. The authors should verify if this is indeed the case.

**Limitations:**

No negative societal impact as this is foundational work that is not considering a specific application or using an datasets.  The actual differential equations being solved are toy problems.

Limitations of the method were addressed by the review in terms of capturing the accuracy of the neural solution.  This limitation was not discussed in the paper.

---

> ### Author Rebuttal · Authors · 2023-08-10
>
> Thanks for your thoughtful feedback. We will address your comments point by point below, and we hope that our answers can ease your concern.
>
> **To Weakness 1:** In terms of not considering epistemic uncertainty, the reviewer is right and we have already clarified in the paper that we consider only proper solvers (see Definition 3.3) and thus, the focus is placed on the propagation of aleatoric uncertainty.
> More specifically, epistemic uncertainty is simplified to be known and fixed as $\eta$.
> As for the specific quantification method for $\eta$, it is beyond the scope of this article's discussion.
> Moreover, any quantification method for $\eta$ can be used as a plug-in with PICProp, without affecting the propagation of aleatoric uncertainty.
>
> We would like to clarify that we do not neglect the significance of epistemic uncertainty. Our primary goal has been to introduce a completely new problem and perspective in UQ that is accompanied by theoretical guarantees.
> This goal has required formulating a new problem, proving theoretical guarantees, comparing with alternative approaches at a theoretical level (see Appendix C), and developing efficient methods for
> solving the resulting bi-level optimization problem.
> We believe that transforming our idea into a general and versatile UQ framework is a process that may require several steps/papers in the near future.
>
> **To Weakness 2:** We thank the reviewer for pointing out this crucial issue. We conducted the following experiment to address the reviewer's concern about the validity of CIs.
>
> For the pedagogical example, we independently sampled 500 boundary conditions and propagate the corresponding Chi-squared CIs using EffiPICProp to construct CIs for $x \in [-1, 1]$.
> According to the Definition of joint CIs (see Definition 3.1), an instance of CI is valid if it contains the solution $u(x)$ for any $x \in [-1, 1]$.
> In this regard, we uniformly sampled 100 data points from $[-1, 1]$ and regard an instance of CI as valid if the solution was enclosed within the interval for these 100 data points.
> Further, we run vanilla PINNs with clean boundary data and use the max absolute error among these 100 data points as $\eta$.
>
> As a result, 486 out of these 500 (0.972 $>$ 0.95) instances of CI are valid, which indicates that our approach to constructing CI is empirically valid and conservative.
>
> Due to the limited time, we will add more results in a later revision.
>
> **To Weakness 3:** We appreciate the valuable suggestions from the reviewers regarding the formatting of the article. Due to space limitations, we have placed Algorithms 1 and 2 in the appendix. However, when reformatting, we will consider moving Algorithms 1 and 2 to the main body of the article.
>
> **To Q1:** The reviewer is right. We will correct this typo in the revised manuscript.
>
> **To Q2:** We assume the reviewer meant "Four lines above eqn. (5)". The reviewer is right. We will correct this typo in the revised manuscript.
>
> **To Q3:** Thanks for the reviewer's suggestion. We use the term "SimPICProp" for the naive regression approach to distinguish it from the meta-learning approach, EffiPICProp ($\lambda > 0$). We will consider the reviewer's suggestion for revision.
>
> **To Q4:** By CI in Def. 3.1, we meant **joint** confidence interval, which is exactly confidence sequence, to the best of the authors' knowledge.
> We use the term (joint) confidence interval as it is commonly used in the literature ([27][29][19]).
> For revision, we will cite the work of Ramdas and point out the equivalence between confidence sequence and joint confidence interval.
> We will also modify Def. 3.1 to avoid ambiguity.

---

> > ### Comment · Reviewer_5z2U · 2023-08-15
> >
> > The authors have addressed my concerns in their rebuttal. I also feel that the authors did a good job addressing the other comments as well.
> >
> > While I do think it would be nice to tackle the total uncertainty including the epistemic uncertainty of the PINN,  I realize that I may have been overly harsh on this point in light of the comments from other reviewers. I do appreciate that the papers does make a contribution that is sound under the $\eta$-proper assumption for the PINN.
> >
> > I have increased the scores to indicated that I am leaning towards acceptance.

---

### Official Review · Reviewer_JUgH · 2023-06-24

**Soundness:** 3 good
**Presentation:** 4 excellent
**Contribution:** 3 good
**Rating:** 7
**Confidence:** 4

**Summary:**

The paper focuses on a novel algorithm for confidence interval estimation for deterministic partial differential equations. It provides leverages bi-level optimization to compute a valid CI, that is supported by both theoretical and numerical investigations, focusing on physics-informed learning.

**Strengths:**

The paper is very well written, with clearly structured sections and appendices. It provides a lot of technical detail and manages to effectively balance what to be included in the main text and what to be pushed to the appendices in order to not distract. I particularly enjoyed reading this work as in blends theoretical aspects with computational ones, and has direct and immediately applicability to the computational science world, particularly error estimations.

The authors have performed a number of benchmarks (of increasing complexity) for the methodology, and have clearly showed its advantages as well as observed ("We conclude that EffiPICProp, although computationally cheaper, in most cases provides wider [...] CIs, as compared to ES with varying numbers of trials") or potential drawbacks ("Although not encountered in the experiments, a potential limitation of our method is that the obtained CIs can be loose in some cases, even when propagating a tight CI from the boundaries, due to the max/min operations in the bi-level optimization of Equation 3").

Overall, I believed the presentation of the work is of high quality and has impact implications to its directed target.

**Weaknesses:**

A weakness of the paper is the fact that some benchmark results are incomplete. Notably SimPICProp in the pedagogical example is only shown for Chi-squared, not for T-squared or Hoeffding. Also SimPICProp and PICProp are not shown for the 2D Poisson equation or the 1D time-dependent Burgers equation.

Lastly, while as the authors say "The aim of this paper is to propose a principled and theoretically guaranteed solution approach and thus, we refrain from applying such ad hoc solutions in the computational experiments and plan to consider parameter uncertainty rigorously in a future work. Nevertheless, we provide in Appendix L a comparison with Bayesian PINNs for completeness.", the figure in Appendix L suggest a potential better performance of the Bayesian PINNs compared to EffiPICProp. For objectivity reasons, I would suggest moving some of that discussion into the main text.

**Questions:**

1. Please complete table J.2 and the show the other comparisons in the figures 3,4,5 of SimPICProp and PICProp to ES and EffiPICProp.
2. In appendix C, you should mentioned the well-known result of Jensen's inequality as the figures are an illustration of this.
3. Please consider discussing the Bayesian PINNs in the main text.


**Limitations:**

Yes

---

> ### Author Rebuttal · Authors · 2023-08-10
>
> Thank you for your thoughtful and overall positive feedback. Here, we provide answers to your concerns point by point.
>
> **To Weakness 1 and Q1**: We thank the reviewer for pointing out the absence of some results. We itemize our response to this question as follows:
>
> (a) We will show SimPICProp results in the pedagogical example for T-squared or Hoeffding CIs in the appendix of the revised manuscript.
>
> (b) SimPICProp is a special case of EffiPICProp when $\lambda$ equals 0.
> We use the term "SimPICProp" for the naive regression approach to distinguish it from the meta-learning approach, EffiPICProp ($\lambda > 0$).
> For 2D Poisson equation and 1D time-dependent Burgers equation, we split $10\%$ of the training data as the validation set to select the best $\lambda$ from $\{0.0, 0.25, 0.5, 0.75, 1.0\}$ (see line 606).
> Notice that $\lambda=0$, i.e. SimPICProp is in the candidate set.
> In this regard, SimPICProp results and costs are not separately reported, as SimPICProp is a special case of the principled EffiPICProp.
> For revision, we will include EffiPICProp results with different $\lambda$s, because we believe such results will help the reader to better understand the power of meta-learning.
>
> (c) The reason we omit the result of (brute-force) PICProp for 2D PDEs is that the cost is too expensive. For example, the grid we use for plotting the result for Burgers equation is $256 \times 100$, which requires $25600$ queries. Each query takes 5000s, as reported in Table J.2. The total computational cost would be 4 GPU years, while EffiPICProp only takes 14 GPU hours.
>
> **To Weakness 2 and Q3:** We thank the reviewer for the revision suggestion. We will accordingly move some of that discussion into the main text in the revised manuscript.
>
> **To Q2:** We thank the reviewer for the constructive suggestion. We will mention Jensen's inequality in the revised manuscript.

---

> > ### Comment · Reviewer_JUgH · 2023-08-10
> >
> > Dear authors,
> >
> > Thank you for the time taken to write the rebuttal and addressing the points I raised.

---

### Official Review · Reviewer_Jho9 · 2023-07-08

**Soundness:** 3 good
**Presentation:** 4 excellent
**Contribution:** 3 good
**Rating:** 7
**Confidence:** 2

**Summary:**

Authors present a method to propagate confidence intervals from some locations to the entire domain. In specific, they consider the problem of propagating uncertainty / confidence internals at the boundary, in partial differential equations based systems, to the rest of the domain. The proposed method PICProp is a bi-level optimization method. Authors present theoretical guarantees for PICProp and then develop a computationally efficient version of the PICProp under some simplifications. The authors show the efficacy of the method via some computational experiments.

**Strengths:**

The paper tackles an important problem of estimating and propagating uncertainty in physics-informed ML systems. The proposed methods are the first methods (to the best of my knowledge) to develop confidence interval propagation method for PDE systems.

The authors present a method, Physics-Informed Confidence Propagation (PICProp), to propagate confidence intervals from any boundary points in the domain to any other points in the domain. They show theoretical guarantees which demonstrate that PICProp propagates confidence intervals with high probability.  The theoretical analysis offers insights into the inner workings of PICProp.

Then authors identify that PICProp is computationally expensive and might not be applicable beyond small domains. To address this they develop an efficient version of PICProp known as EffiPICProp which makes confidence interval propagation much more computationally viable. It is great that authors through about the practicality of the algorithm and developed this.

Authors test their method on computational experiments. In spite of being small scale experiments, they offer intuition which is important for transferring these methods to other domains / problems. It is great that the authors have actually documented the computational time for running different methods on these problems.



**Weaknesses:**

All the computational experiments seems to be conducted on 2D domains. It is not clear how the computational costs scale once we go to higher dimensions and if EffiPICProp can still be computationally feasible. It might be useful to discuss about this.

This work deals with deterministic systems and often most real-life applications involve stochastic systems (for example, due to inaccurate modeling of the systems). It might be useful to discuss about how the proposed methods can or cannot transfer to stochastic systems.

As of now, the proposed method only propagates uncertainty from boundary points.

**Questions:**

It would be great if authors can comment about points mentioned in "Weakness" section

**Limitations:**

Please refer to "Weakness" section.

Even though the points mentioned in the weakness section can be thought of as limitations, I don't find them to be major.

---

> ### Author Rebuttal · Authors · 2023-08-10
>
> Thank you for your thoughtful and overall positive feedback. Here, we provide answers to your concerns point by point.
>
> **To Weakness 1:** The dimensionality of PDE/ODE problems is typically N+1, where N is the number of spatial dimensions and 1 accounts for the temporal dimension. In this regard, the problems that we considered in this paper involve standard representative equations met in many real-world problems including heat transfer, fluid dynamics, flows through porous media, and even epidemiology [1-2] and N is in many practical cases equal to 1 or 2. The problems that we selected to present in this paper are similar to or the same as the standard problems solved by the computational science community for developing novel solution methods (both deterministic [3] and considering uncertainty [4]).
>
> [1] Meng, X., Babaee, H. and Karniadakis, G.E., 2021. Multi-fidelity Bayesian neural networks: Algorithms and applications. Journal of Computational Physics, 438, p.110361.
>
> [2] Zou, Z., Meng, X., Psaros, A.F. and Karniadakis, G.E., 2022. NeuralUQ: A comprehensive library for uncertainty quantification in neural differential equations and operators. arXiv preprint arXiv:2208.11866.
>
> [3] Psaros, A.F., Kawaguchi, K. and Karniadakis, G.E., 2022. Meta-learning PINN loss functions. Journal of computational physics, 458, p.111121.
>
> [4] Yang, Y. and Perdikaris, P., 2019. Adversarial uncertainty quantification in physics-informed neural networks. Journal of Computational Physics, 394, pp.136-152.
>
> **To Weakness 2:** We thank the reviewer for the constructive suggestion.
> The solution of a stochastic system is itself stochastic.
> In this regard, confidence intervals are no longer applicable, as confidence intervals are defined for unknown constant statistics rather than random variables.
> Although the problem formulation of uncertainty quantification for stochastic systems is totally different, the idea to propagate the uncertainty might be adapted.
> For example, if we can construct CIs for the force terms, we can propagate the uncertainty from force terms to the solutions by including the force terms into the bi-level optimization in the exact same way as the boundary conditions.
> We leave the more detailed problem formulation, theories, and algorithms for future work.
>
> **To Weakness 3:** PICProp can be easily generalized to propagate uncertainty from inner data points or collocation data points, once CIs/data to construct CIs at these points are available. We plan to work on it in the future.

---

> > ### Comment · Reviewer_Jho9 · 2023-08-14
> >
> > Thanks for taking time and addressing the points I have raised.

---

### Official Review · Reviewer_Lv8K · 2023-07-09

**Soundness:** 2 fair
**Presentation:** 2 fair
**Contribution:** 2 fair
**Rating:** 4
**Confidence:** 4

**Summary:**

The paper discusses the limitations of current approaches for uncertainty quantification in deep learning and physics-informed learning, such as the need for strong assumptions regarding data likelihood and the computational cost associated with approximating the posterior. The paper introduces a novel problem of confidence interval estimation for deterministic partial differential equations and proposes a method called Physics-Informed Confidence Propagation (PICProp) based on bi-level optimization to compute a valid confidence interval without making heavy assumptions. The paper provides a theorem regarding the validity of the method and computational experiments focusing on physics-informed learning.

**Strengths:**

1. The paper is simple and clear to follow
2. The paper is well structured
3. The paper aims at solving an important problem involving UQ in PINN
4. The theoretical analysis is helpful to better understand the benefits of the method

**Weaknesses:**

1. The paper claims efficiency but did not demonstrate the cost saving with enough supports
2. The scope is narrowed to PDE but did not see practical applications, e.g., large-scale cases
3. The experiments only focus on toy examples that did not show the strength of this work.

**Questions:**

Similar to the weakness, I have several concerns about the claims and experiments demonstration

1. The experiments can be solved by some baseline methods as the author mentioned because the cost is acceptable. If the authors would like to show the real power of this method, they may consider some real-world problems, either from PDE or simulations.

2. The computational cost is the major barrier to the UQ problem. The method leverages bi-level optimization, how about the cost compared with the traditional methods, e.g., MC-based approaches, specifically in complex cases? For example, high-fidelity simulations, CDF or FEM, can you handle the optimization efficiently without any approximations?  How about the computational cost?

3. I only saw the analytical experiments and have been thinking about how to apply this method to solve real-world applications, is that useful or helpful to handle complex cases in AI for science problems?

----
After rebuttal

I posed several follow-up questions but unfortunately, I did not receive the authors' positive responses although they made efforts to argue. As a result, I incline to keep my original score.

**Limitations:**

Did not see a clear discussion about the limitations.

---

> ### Author Rebuttal · Authors · 2023-08-10
>
> Thank you for your thoughtful feedback. Here, we provide answers to your concerns point by point.
>
> **To Q1:** We assume that the reviewers are referring to the exhaustive search (ES) method as the baseline, of which the cost is acceptable.
>
> We would like to clarify that the cost of ES is acceptable only for the pedagogical example (2 boundary data points).
> While acceptable, the cost of ES (42 hrs for 5k trials) is several times higher than that of EffiPICProp (less than 20 mins) to achieve a similar approximation of the solution of Eq(4).
> As the number of data points increases, e.g. 2D Poisson equation (40 boundary data points) and Burgers equation (256 boundary points), the cost is not acceptable anymore.
> Even with a cost budget several times larger than that of PICProp, ES still struggles to well approximate the solution of Eq(4) accurately. One possible reason is that as the number of data points increases, the search space grows exponentially, leading to a corresponding exponential increase in search costs.
> In this regard, the authors conclude that PICProp is more powerful and scalable compared to ES, in terms of solving Eq(4).
>
> If the reviewer meant traditional UQ approaches as baselines, we would like to direct the reviewer to Appendix B, C, and L for more detailed discussions about the difference between PICProp and traditional UQ methods (e.g. Bayesian method, MC simulation), especially the differences in the problems they address and the properties of the probabilistic intervals they construct. Overall, the results obtained by these methods can be significantly different both in terms of interpretation and quantitatively. In this regard, these approaches cannot serve as baselines for comparison with our approach.
>
> Moreover, the problems that we considered in this paper actually met in many real-world problems. Please also refer to the answer to Q3.
>
> **To Q2:** We itemize our response to this question as follows:
>
> (a) PINNs as a deep learning approach for solving PDEs are an alternative to conventional CDF or FEM simulations.
> As the focus of this paper is to use PINNs as the underlying numerical solver of the PDE to construct CIs, the comparison between PINNs with traditional simulation approaches is outside the scope of this paper and has been discussed in [1].
>
> [1] Karniadakis G E, Kevrekidis I G, Lu L, et al. Physics-informed machine learning[J]. Nature Reviews Physics, 2021, 3(6): 422-440.
>
> (b) A seemingly feasible alternative to our approach is Monte Carlo simulation using multiple FEM simulations. However, it has the same limitations as Monte Carlo simulation using PINNs as the PDE solver, which are discussed in Appendix C.
>
> (c) We would like to emphasize that there are many differences between PICProp and traditional UQ methods (e.g. Bayesian method, MC simulation), as discussed in Appendix B, C, and L.
> Overall, the results obtained by these methods can be significantly different both in terms of interpretation and quantitatively.
> As a result, we decided to omit such a comparison for avoiding any potential confusion of the readers.
>
> (d) Gradient-based approaches used in this work for bi-level optimization, are all approximate methods.
> Please refer to Appendix I for more details on approximating the implicit gradient.
> In terms of computational cost, all three methods for approximating implicit gradient used in this paper provide hyperparameters to balance computational cost and accuracy.
> The performance of these methods depends on the specific problem and is difficult to analyze.
> For the problems studied in this paper, we summarize the empirical computational cost in Table J.2.
> Compared to the search-based method, bi-level optimization-based methods
> are more efficient.
>
> **To Q3:** The dimensionality of PDE/ODE problems is typically N+1, where N is the number of spatial dimensions and 1 accounts for the temporal dimension. In this regard, the problems that we considered in this paper involve standard representative equations met in many real-world problems including heat transfer, fluid dynamics, flows through porous media, and even epidemiology [1-2] and N is in many practical cases equal to 1 or 2. The problems that we selected to present in this paper are similar to or the same as the standard problems solved by the computational science community for developing novel solution methods (both deterministic [3] and considering uncertainty [4]).
>
> [1] Meng, X., Babaee, H. and Karniadakis, G.E., 2021. Multi-fidelity Bayesian neural networks: Algorithms and applications. Journal of Computational Physics, 438, p.110361.
>
> [2] Zou, Z., Meng, X., Psaros, A.F. and Karniadakis, G.E., 2022. NeuralUQ: A comprehensive library for uncertainty quantification in neural differential equations and operators. arXiv preprint arXiv:2208.11866.
>
> [3] Psaros, A.F., Kawaguchi, K. and Karniadakis, G.E., 2022. Meta-learning PINN loss functions. Journal of computational physics, 458, p.111121.
>
> [4] Yang, Y. and Perdikaris, P., 2019. Adversarial uncertainty quantification in physics-informed neural networks. Journal of Computational Physics, 394, pp.136-152.

---

> > ### Comment · Reviewer_Lv8K · 2023-08-21
> > **Thanks for your response!**
> >
> > Dear author team,
> >
> > Thanks very much for your response.  Some of my concerns are addressed but some core questions are still ignored, specifically more experiments to demonstrate the practical applicability of this method.
> >
> > The authors repeat again about the potential real-world applications, but they did not show any experiments. We understand PDE is the foundation of many math and physics problems, but why do we care about real-world applications e.g., AI for drugs, rather than only showing PDE or toy examples? Many complex physics challenges (e.g., climate changes) can be not well solved by only exploring PDE, isn't it?
> >
> > Can we say that we solve the PDE problems well and then our method can solve all PDE-based real-world applications?  If so, do we need to collect real-world data, and figure out details in real-world applications?  If you think the answer is yes, maybe the better choice is SIAM or math journal/conferences, which provide more foundational research with more solid theoretical studies?
> >
> > Although the authors show the promise of this method, I still doubt that the proposed method is ONLY limited to PDE or toy examples without any additional experiments on real-world applications.   Thus the existing work can not convince me this method is meaningful or useful for solving real-world applications. In contrast, I only note that many references and potential discussions from this rebuttal but without extensive efforts/motivations on dealing with real-world applications.

---

> > > ### Author Response · Authors · 2023-08-21
> > >
> > > Dear reviewer,
> > >
> > > We thank the reviewer for the comment and we would like to add some clarifications to our previous comment. Please note that the cases that we considered in the paper can already be viewed as real-world applications. For example, the Burgers equation is often used as a simplified version of the Navier-Stokes equation for fluid mechanics applications.
> > >
> > > The significance of PINNs and more generally AI as forward and inverse problem solvers has been discussed in detail in the references posted in our previous comment. In this paper, we take the significance of AI for science for granted and focus on UQ. Specifically, in the paper we have stressed the importance of UQ and showed the limitations of other UQ approaches using PINNs. Subsequently, we proceeded with our method and solved benchmark problems as typically done in papers published by the computational science community. In doing so, we have considered multiple real-world data availability scenarios.
> > >
> > > Additionally, the contributions of our work encompass the formulation of the CI estimation problem for PDEs, a comprehensive discussion on the similarities, differences, strengths, and weaknesses compared to standard UQ problems, algorithm design and theoretical analysis, as well as experiments conducted on standard benchmarks. These components align with the established standards of active fundamental research on PINNs. We respectfully disagree that not being immediately applied to real-world physical challenges like climate change should be viewed as a main drawback of this work.

---

> ### Author Response · Authors · 2023-08-16
>
> We want to gently ping here to see if our answers have addressed your concerns. We sincerely hope that we can provide answers to ease your concerns and would love to try again if you think our current answers are not satisfying,

---

> ### Comment · Area_Chair_MAxY · 2023-08-20
>
> To Reviewer Lv8K:
>
> You should respond at least once to the authors' rebuttal.
> Your score is lower than those of the other reviewers, so your opinion is important for final judgment.
> Please read the other reviewers' comments and the author's rebuttal and update your final score if necessary.
>
> Thanks for your cooperation.

---

### Official Review · Reviewer_H2bG · 2023-07-12

**Soundness:** 3 good
**Presentation:** 3 good
**Contribution:** 3 good
**Rating:** 6
**Confidence:** 3

**Summary:**

The paper introduces a PICProp, a method to propagate uncertainty through spatiotemporal nonlinear PDE systems. PICProp (1) is more valid and makes less assumptions compared to NN based UQ methods such as dropout and deep ensembles, and (2) can be modified to be more computationally efficient by learning a meta-model. The paper provides extensive computational to show the effectiveness of PICProp.

**Strengths:**

- The methodological section is very clear and easy to follow. PICProp is a principled, concise, and elegant method for the proposed problem setup.
- I really enjoy the writing and organization in the experiment section as well. The pedagogical example is illustrative of how the algorithm works for different types of CIs, and the following sections showcase the effectiveness of the method in 2d and in PDEs with steep gradients.

**Weaknesses:**

1. The aper mentions that PICProp produces conservative CIs that are "clearly preferred in critical and risk sensitive applications". I don't think this statement is necessarily true. In many cases, one wants the CIs to be valid and **informative**. You can have very large CIs that are trivially valid but too wide to have any meaning in decision-making. (In general, wide CI = safety is neither a correct or scientifically sound statement, don't use it in your papers. Line 285 is an example.) There exist metrics such as Winkler Score and interval scores [1] that principally evaluate the quality of a CI; I recommend the authors to use them. The authors did mention this shortcoming in the conclusion section and say that they will study it in a future work - My opinion is that these studies need to be a part of *this* work for the method to be convincing and usable for the community.

[1] Gneiting, Tilmann, and Adrian E. Raftery. "Strictly proper scoring rules, prediction, and estimation." Journal of the American statistical Association 102.477 (2007): 359-378.

2. The marginal probability framing of CIs (Theorem 4.1) is very close to the guarantees provided by frequentist UQ methods (jackknife and conformal prediction, for example). These are also "standard", distribution-free, and widely used UQ methods; it's important to mention and compare to them. Comparing to them will likely help you prove your case of why it's important to have CIs be physically informed.

3. This might be more from of a machine learning community perspective, but I'm curious to see PICProp's performance on a real dataset instead of just on synthetic experiments. The applications authors mentioned in appendix A are all good candidates.



**Questions:**

See weaknesses.

---

> ### Author Rebuttal · Authors · 2023-08-10
>
> Thank you for your thoughtful feedback. Here, we provide answers to your concerns point by point.
>
> **To Weakness 1:** We would like to clarify that none of the statements in our paper are intended to give readers the impression that "wider CI = safety." In fact, in practical applications, constructing confidence intervals involves a trade-off between practicability and safety.
> Typically, we aim for informative and valid confidence intervals to aid in making more accurate decisions. However, in **risk-sensitive** applications, considering the significant harm of non-valid confidence intervals, constructing conservative confidence intervals by sacrificing some information to ensure the validity might be preferable.
>
> Given that commonly used uncertainty quantification (UQ) methods, including deep ensembles and MC dropout, tend to be over-confident (too narrow to be valid) in practical applications, we emphasize the conservatism of PICProp as a distinguishing property from traditional approaches. Users can select accordingly based on the specific requirements of their practical applications.
>
> Regarding revision, we will follow the reviewer's suggestions to (1) enhance the comprehensiveness of the relevant discussions in the introduction and (2) revise other parts of the manuscript where there might be statements that inadvertently suggest "Wide CI = safety," including but not limited to line 285, line 309 and the caption of Figure 5.
>
> We agree with the reviewer that universal metrics for evaluating confidence intervals will help to enhance the comprehensiveness of this work. However, interval scores defined in [1] are for **prediction intervals** rather than confidence intervals. To the best of the authors' knowledge, such a metric for confidence intervals has not been well studied yet.
>
> We are open to incorporating general metrics to evaluate confidence intervals in future research. For example, a straight-forward generalization of interval scores to confidence intervals may be $S(L, U, z) = U(z) - L(z) + \frac{\alpha}{2}(L(z) - u(x))\mathbb{I}\{u(x) < L(z)\} + \frac{\alpha}{2}(u(x) - U(z))\mathbb{I}\{u(x) > U(z)\}$, where $z$ denotes an instance of boundary data, and $(L(x), U(x))$ denotes the constructed CI for the solution $u(x)$ at $x$. But still, these potential metrics need to be thoroughly studied before being applied to our work.
>
> **To Weakness 2:** We would like to highlight that the problem studied in this work is to construct **confidence intervals** for the solution of PDEs, which is constant as we are considering deterministic systems. Jackknife and conformal prediction mentioned by the reviewer, are approaches to construct \textbf{prediction intervals} for the next sample, which is indeed a random variable. Thus, although considered as standard UQ methods, they cannot be directly applied to construct CIs.
>
> We also would like to direct the reviewer to Appendix B, C, and L for more detailed discussions about the difference between PICProp and traditional UQ methods (e.g. Bayesian method, MC simulation), especially the differences in the problems they address and the properties of the probabilistic intervals they construct. Overall, the results obtained by these methods can be significantly different both in terms of interpretation and quantitatively.
> As a result, we decided to omit such a comparison for avoiding any potential confusion of the readers.
>
> **To Weakness 3:** The dimensionality of PDE/ODE problems is typically N+1, where N is the number of spatial dimensions and 1 accounts for the temporal dimension. In this regard, the problems that we considered in this paper involve standard representative equations met in many real-world problems including heat transfer, fluid dynamics, flows through porous media, and even epidemiology [1-2] and N is in many practical cases equal to 1 or 2. The problems that we selected to present in this paper are similar to or the same as the standard problems solved by the computational science community for developing novel solution methods (both deterministic [3] and considering uncertainty [4]).
>
> [1] Meng, X., Babaee, H. and Karniadakis, G.E., 2021. Multi-fidelity Bayesian neural networks: Algorithms and applications. Journal of Computational Physics, 438, p.110361.
>
> [2] Zou, Z., Meng, X., Psaros, A.F. and Karniadakis, G.E., 2022. NeuralUQ: A comprehensive library for uncertainty quantification in neural differential equations and operators. arXiv preprint arXiv:2208.11866.
>
> [3] Psaros, A.F., Kawaguchi, K. and Karniadakis, G.E., 2022. Meta-learning PINN loss functions. Journal of computational physics, 458, p.111121.
>
> [4] Yang, Y. and Perdikaris, P., 2019. Adversarial uncertainty quantification in physics-informed neural networks. Journal of Computational Physics, 394, pp.136-152.

---

> > ### Comment · Reviewer_H2bG · 2023-08-14
> > **Increased score**
> >
> > Apologies for the mix up between confidence interval vs. prediction intervals, thanks for the clarification.
> >
> > Given the author's proposed revisions, I have increased my score.

---

### Decision · Program_Chairs · 2023-09-21

**Decision:**

Accept (poster)

**Comment:**

This paper addresses the important problem of uncertainty estimation and propagation in physically informed ML systems. The authors proposed a confidence interval propagation method for PDE systems. Specifically,  this paper proposes a method called Physically Informed Confidence Interval Propagation (PICProp), which propagates a confidence interval from any boundary point in a domain to any other point in the domain, and conducts a theoretical analysis. They further show that PICProp is computationally expensive and may not be applicable outside of small regions, and to address this, the authors develop an efficient version of PICProp known as EffiPICProp that makes confidence interval propagation more computationally feasible. The authors validate their method through computational experiments.